# On the Gini-impurity Preservation
# For Privacy Random Forests

**Xin-Ran Xie**,* **Man-Jie Yuan**,* **Xue-Tong Bai, Wei Gao, Zhi-Hua Zhou**
National Key Laboratory for Novel Software Technology, Nanjing University, China
School of Artificial Intelligence, Nanjing University, China
`{xiexr,yuanmj,baixt,gaow,zhouzh}@lamda.nju.edu.cn`

## Abstract

Random forests have been one of the successful ensemble algorithms in machine learning. Various techniques have been utilized to preserve the privacy of random forests, such as anonymization, differential privacy, homomorphic encryption, etc. This work takes one step towards data encryption by incorporating some crucial ingredients of learning algorithm. Specifically, we develop a new encryption to preserve data's Gini impurity, which plays an important role during the construction of random forests. The basic idea is to modify the structure of binary search tree to store several examples in each node, and encrypt the data features by incorporating label and order information. Theoretically, our scheme is proven to preserve the minimum Gini impurity in ciphertexts without decrypting, and we also present the security guarantee for encryption. For random forests, we encrypt data features based on our Gini-impurity-preserving scheme, and take the homomorphic encryption scheme CKKS to encrypt data labels owing to their importance and privacy. We finally present extensive empirical studies to validate the effectiveness, efficiency and security of our proposed method.

## 1   Introduction

From the pioneer work [1], random forests have been one successful ensemble algorithm [2–4], with diverse applications such as ecology [5], computational biology [6], objection recognition [7], remote sensing [8], computer vision [9], etc. The basic idea is to construct a large number of random trees individually and make prediction based on an average of their predictions. Numerous variants of random forests have been developed to improve performance under different settings [10–22], as well as theoretical understandings on the success of random forests [21, 23–27]. The splitting criterion, such as Gini impurity and information gain, has been one of the most important ingredient during the construction of random forests [1, 28].

Various techniques have been adopted to preserve the privacy of random forests, especially for sensitive tasks such as medical diagnosis, financial predictions, and so on. For example, differential privacy [29] has been successfully applied to preserve the privacy of random forests [30, 31] and decision trees [32–34], by adding certain noise perturbations. Another relevant approach is the secure multi-party computation for random forests and decision tree [35–39], where the privacy is preserved by multi-party joint computation over respective data inputs without leakage.

Homomorphic encryption [40–43] has been one of the most important cryptosystems in privacy-preserving computing [44–47]. Based on such scheme, various algorithms have been developed to train privacy random forests and decision trees [48–52], while some other methods only considered inference without training due to computational costs [53–58]. In addition, LeFevre et al. [59] took

---

*These authors contribute equally.

37th Conference on Neural Information Processing Systems (NeurIPS 2023).

**Table 1:** Comparisons of communications and complexities for different privacy-preserving decision trees. Here, $n$ is the number of examples in training data, and $\tau$ is the cardinality of label space. Let $h$ and $\kappa$ be the height and number of leaves of decision tree ($h < \kappa$), respectively. Denote by $\bar{\jmath}$ the average number of possible splitting features and positions in the construction of decision trees, and $p$ is the number of clients for secure multi-party computation. '–' means the corresponding methods focusing only on inference without training.

| Scheme | Training communication | | Training comp. complexity | | Predictive communication | | Predictive comp. complexity | | Privacy of model |
|---|---|---|---|---|---|---|---|---|---|
| | Rounds | Bandwidth | Client | Server | Rounds | Bandwidth | Client | Server | |
| SMCDT [61] | $O(\kappa)$ | $O(\bar{\jmath}\tau n)$ | $O(\kappa\bar{\jmath}\tau n)$ | $O(\kappa\bar{\jmath}\tau)$ | $O(1)$ | $O(1)$ | $O(1)$ | $O(h)$ | ✗ |
| PPID3 [36] | $O(\kappa)$ | $O(p^2\bar{\jmath}\tau n)$ | $O(\kappa p^2\bar{\jmath}\tau n)$ | $O(\kappa p^2\bar{\jmath}\tau n)$ | $O(1)$ | $O(1)$ | $O(1)$ | $O(h)$ | ✗ |
| SID3 [37] | $O(hp)$ | $O(\kappa\bar{\jmath}\tau)$ | $O(\kappa\bar{\jmath}\tau n)$ | $O(\kappa\bar{\jmath}\tau)$ | $O(1)$ | $O(1)$ | $O(1)$ | $O(h)$ | ✗ |
| OPPC4.5 [39] | $O(\kappa p)$ | $O(p\bar{\jmath}\tau)$ | $O(\kappa\bar{\jmath}\tau(n+p))$ | $O(m\bar{\jmath}\tau p)$ | $O(1)$ | $O(1)$ | $O(1)$ | $O(h)$ | ✗ |
| PivotRFs [62] | $O(\kappa p)$ | $O(\bar{\jmath}\tau + \tau n)$ | $O(\kappa\bar{\jmath}\tau n)$ | $O(\kappa\bar{\jmath}\tau)$ | $O(p)$ | $O(\kappa)$ | $O(\kappa)$ | $O(\kappa)$ | ✓ |
| MulPRFs [63] | $O(h)$ | $O(\log n + \log d)$ | $O(hdn\log n)$ | $O(hdn\log n)$ | $O(h)$ | $O(1)$ | $O(h)$ | $O(h)$ | ✗ |
| PPD-ERTs [64] | $O(hp)$ | $O(\kappa\bar{\jmath}\tau)$ | $O(\kappa\bar{\jmath}\tau n)$ | $O(\kappa\bar{\jmath}\tau)$ | $O(1)$ | $O(1)$ | $O(1)$ | $O(h)$ | ✗ |
| HEldpRFs [51] | $O(h)$ | $O(\kappa\bar{\jmath}\tau)$ | $O(\kappa\bar{\jmath}\tau)$ | $O(\kappa\bar{\jmath}\tau n)$ | $O(1)$ | $O(\kappa)$ | $O(\kappa)$ | $O(\kappa)$ | ✓ |
| SecureDT [65] | – | – | – | – | $O(1)$ | $O(1)$ | $O(1)$ | $O(\kappa)$ | ✓ |
| PrivateDT [66] | – | – | – | – | $O(1)$ | $O(1)$ | $O(1)$ | $O(\kappa)$ | ✓ |
| Our work | $O(h)$ | $O(\kappa\bar{\jmath})$ | $O(\kappa)$ | $O(\kappa\bar{\jmath}\tau n)$ | $O(1)$ | $O(1)$ | $O(1)$ | $O(h)$ | ✓ |

the anonymization [60] for random forests by grouping similar attributes so as to hardly identify specific individual information.

This work takes one step towards data encryption by incorporating some crucial ingredients of learning algorithm, and main contributions can be summarized as follows:

- We present a new encryption to preserve data's Gini impurity, and the basic idea is to modify the structure of binary search trees to maintain several samples on each node, and encrypt data's features by incorporating label and order information. Our scheme could change the data frequencies, which is also beneficial for data security.

- Theoretically, we prove the preservation of minimum Gini impurity in ciphertexts without decryption, which plays an important role on the construction of random forests. Our scheme also satisfies the security against Gini-impurity-preserving chosen plaintext attack.

- We focus on the privacy random forests in the popular client-server protocol, and take our Gini-impurity-preserving encryption for data features. We adopt homomorphic encryption CKKS to encrypt data labels. Our encrypted decision tree takes smaller communication and computational complexities, as shown in Table 1.

- Extensive experiments show that our encrypted random forests take significantly better performance than prior privacy random forests via encryption, anonymization and differential privacy, and are comparable to original (plaintexts) random forests without encryption. Our encrypted random forests make a good balance between computational cost and data security.

The rest of this work is constructed as follows: Section 2 introduces relevant work. Section 3 presents an encryption on data's Gini impurity. Section 4 proposes the encrypted random forests. Section 5 conducts extensive experiments. Section 6 concludes with future work.

## 2 Relevant Work

Homomorphic Encryption (HE) is a cryptosystem, which allows operations on encrypted data without access to a secret key [40]. We can perform some mathematical operations such as addition and multiplication operations on encrypted data without revealing sensitive information. Given an encryption function $E(\cdot)$ and a decryption function $D(\cdot)$, the HE scheme provides two operators $\oplus$ and $\otimes$ such that, for every pair of plaintexts $x_1$ and $x_2$,

$$D\left(E(x_1) \oplus E(x_2)\right) = x_1 + x_2 \quad \text{and} \quad D\left(E(x_1) \otimes E(x_2)\right) = x_1 \times x_2 \,,$$

where $+$ and $\times$ denote standard addition and multiplication operations, respectively.

Various HE schemes have been developed during the past years, e.g., ElGamal [67], Paillier [68], CKKS [42] encryption, etc. Relevant techniques have been successfully applied to machine learning tasks such as regression problem [69, 70], neural network [71–75], collaborative filtering [76], etc. Generally, HE schemes are accompanied with high computational costs, and one main challenge is to maintain a good trade-off among security, effectiveness and computational cost in real applications.

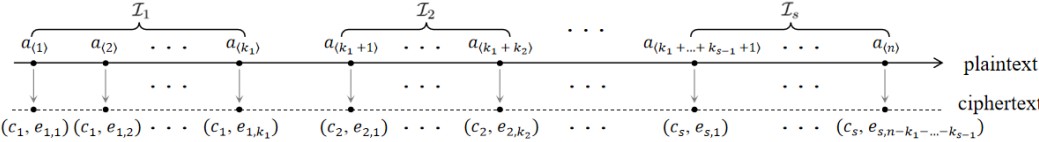

**Figure 1:** A simple illustration for our encryption: each plaintext is encrypted into a ciphertext vector $(c_i, e_{i,j})$. Here, random numbers $c_1 < c_2 < \cdots < c_s$ are introduced to preserve the Gini impurity for random forests, and we take homomorphic encryption scheme for $e_{i,j} = \text{Enc}(k_{\text{pub}}, j)$ in Eqn. (5), which is helpful for decryption.

Secure Multi-Party Computation (SMC) [77] is another cryptographic technique to jointly compute a function from multiple private inputs with confidential, which has been used for machine learning to protect privacy data, such as neural network [78–80], $k$-means clustering [81–83], random forests and decision trees [35–39], etc. Differential privacy is introduced to preserve individual privacy by taking statistically inconsequential changes to data [84], and relevant techniques have been utilized in neural network [85–87], random forests [30, 31] and decision trees [32–34].

We introduce some notations used in this work. Write $[\tau] = \{1, 2, \cdots, \tau\}$ for integer $\tau \geq 2$. Let $\mathcal{X} \subset \mathbb{R}^d$ and $\mathcal{Y} = [\tau]$ denote the feature and label space, respectively. A training sample is given by $S_n = \{(\boldsymbol{x}_1, y_1), (\boldsymbol{x}_2, y_2), ..., (\boldsymbol{x}_n, y_n)\}$. Let $|A|$ be the cardinality of set $A$, and $[\![\cdot]\!]$ denotes the corresponding encrypted value. Let $\mathcal{N}(\mu, \sigma^2)$ be a normal distribution of mean $\mu$ and variance $\sigma^2$.

## 3 An Encryption for Gini Impurity

This section presents the first encryption to preserve the minimum Gini impurity over encrypted data without decryption. For simplicity, we give the detailed encryption on one-dimensional feature by incorporating label information, and make similar considerations for other dimensions.

### 3.1 Theoretical Analysis for Gini Impurity

Let $A = \{(a_1, y_1), \cdots, (a_n, y_n)\}$ be a dataset with labels $y_i \in [\tau]$, and define the Gini value as

$$\text{Gini}(A) = 1 - \sum\nolimits_{y \in [\tau]} p_y^2 \,,$$

where $p_y$ denotes the proportion of the label $y$. Let $A_a^l = \{(a_i, y_i) : a_i \leq a, (a_i, y_i) \in A\}$ and $A_a^r = \{(a_i, y_i) : a_i > a, (a_i, y_i) \in A\}$ be the left and right subsets of $A$ w.r.t. a splitting point $a$, respectively. We define the Gini impurity w.r.t. dataset $A$ and splitting point $a$ as

$$I_G(A, a) = w_l \cdot \text{Gini}(A_a^l) + w_r \cdot \text{Gini}(A_a^r) \,, \tag{1}$$

where $w_l = |A_a^l|/n$ and $w_r = |A_a^r|/n$. Let $I_G^*(A)$ be the minimum Gini impurity of dataset $A$, i.e.,

$$I_G^*(A) = \min_{a \in \mathbb{R}} \{I_G(A, a)\} \,. \tag{2}$$

The minimum Gini impurity plays a crucial role on nodes splitting during the construction of random forests. We re-sort dataset $A$ with a non-decreasing order for $a_1, a_2, \cdots, a_n$ as follows:

$$A = \left\{ (a_{\langle 1 \rangle}, y_{\langle 1 \rangle}), (a_{\langle 2 \rangle}, y_{\langle 2 \rangle}), \cdots, (a_{\langle n \rangle}, y_{\langle n \rangle}) \right\} \,, \tag{3}$$

where $a_{\langle 1 \rangle} \leq a_{\langle 2 \rangle} \leq \cdots \leq a_{\langle n \rangle}$, and $y_{\langle 1 \rangle}, y_{\langle 2 \rangle}, \cdots, y_{\langle n \rangle}$ denote their corresponding labels. By incorporating label information, we partition dataset $A$ into several datasets $\mathcal{I}_1, \mathcal{I}_2, \cdots, \mathcal{I}_s$ as follows:

$$
\begin{aligned}
\mathcal{I}_1 &= \left\{ (a_{\langle 1 \rangle}, y_{\langle 1 \rangle}), \cdots, (a_{\langle k_1 \rangle}, y_{\langle k_1 \rangle}) \right\} \,, \\
\mathcal{I}_2 &= \left\{ (a_{\langle k_1+1 \rangle}, y_{\langle k_1+1 \rangle}), , \cdots, (a_{\langle k_1+k_2 \rangle}, y_{\langle k_1+k_2 \rangle}) \right\} \,, \\
&\cdots \\
\mathcal{I}_s &= \left\{ (a_{\langle k_1+k_2+\cdots+k_{s-1}+1 \rangle}, y_{\langle k_1+k_2+\cdots+k_{s-1}+1 \rangle}), \cdots, (a_{\langle n \rangle}, y_{\langle n \rangle}) \right\} \,.
\end{aligned} \tag{4}
$$

Here, any two adjacent datasets have different labels, and all samples have an identical label in one dataset $\mathcal{I}_j$, i.e., $y_{\langle i \rangle} = y_{\langle i' \rangle}$ for every $(a_{\langle i \rangle}, y_{\langle i \rangle}) \in \mathcal{I}_j$ and $(a_{\langle i' \rangle}, y_{\langle i' \rangle}) \in \mathcal{I}_j$.

---

**Algorithm 1** The Gini-impurity-preserving encryption

---

**Input:** Dataset $A = \{(a_1, y_1), \cdots, (a_n, y_n)\}$
**Output:** Binary search tree $\mathcal{BT}$, ciphertexts $\{[\![a_1]\!], \cdots, [\![a_n]\!]\}$
**Initialize:** Tree $\mathcal{BT} = \emptyset$ with its $cipher_1 = c_{\max}/2$, where $c_{\max} = 2^{\lambda \log_2 n}$

  **for** $i = 1, \cdots, n$ **do**
  %% Step-I: Search a node for sample $(a_i, y_i)$ in binary search tree $\mathcal{BT}$
    Set $t$ = root of $\mathcal{BT}$, $t_{\min} = 0$, $t_{\max} = c_{\max}$ and index = 1
    **while** $t$ is an internal node **and** index==1 **do**
      index= 0
      **if** $t.left \neq \emptyset$ **and** $a_i < \max\{a_j : (a_j, y_j) \in t.left.samples\}$ **then**
        $t = t.left$, $t_{\max} = t.cipher_1$, index = 1
      **else if** $t.right \neq \emptyset$ **and** $a_i > \min\{a_j : (a_j, y_j) \in t.right.samples\}$ **then**
        $t = t.right$, $t_{\min} = t.cipher_1$, index = 1
      **end if**
    **end while**
    Update $t = t.left$ if Eqn. (6) is true, and update $t = t.right$ if Eqn. (7) is true

  %% Step-II: Update the binary search tree $\mathcal{BT}$
    **if** $y_i \neq y_j$ for some $(a_j, y_j) \in t.samples$ **then**
      Split node $t$ by Algorithm 2 with inputs of $(a_i, y_i)$ and the corresponding interval $[t_{\min}, t_{\max}]$
    **end if**
    Append example $(a_i, y_i)$ into $t.samples$ and update $t.cipher_2 = \text{Enc}(k_{\text{pub}}, |t.samples|)$
    Encrypt $[\![a_i]\!] = (t.cipher_1, t.cipher_2)$
  **end for**

---

We consider two important factors in encryption: i) preservation of the minimum Gini impurity $I_G^*(A)$ over the encrypted data, and ii) a cryptosystem for encoding and decoding data. Based on such recognition, we introduce the following encryption, for every example $(a_{\langle i \rangle}, y_{\langle i \rangle}) \in \mathcal{I}_j$,

$$[\![a_{\langle i \rangle}]\!] = ([\![a_{\langle i \rangle}]\!]_1, [\![a_{\langle i \rangle}]\!]_2) = \begin{cases} (c_1, \text{Enc}(k_{\text{pub}}, i)) & \text{for } j = 1, \\ (c_j, \text{Enc}(k_{\text{pub}}, i - k_1 - \cdots - k_{j-1})) & \text{for } 2 \leq j \leq s. \end{cases} \tag{5}$$

Here, $c_1, c_2, \cdots, c_s$ are random numbers s.t. $c_1 < c_2 < \cdots < c_s$, which aim to preserve the minimum Gini impurity. We take the homomorphic encryption scheme CKKS with a public key $k_{\text{pub}}$ for $[\![a_{\langle i \rangle}]\!]_2 = \text{Enc}(k_{\text{pub}}, i - k_1 - \cdots - k_{j-1})$ in Eqn. (5), and it is useful for decryption. Figure 1 presents a simple illustration for our encryption, and the detailed decryption is given in Appendix A.

We now present our main theorem as follows:

**Theorem 1.** *We have $I_G^*(A) = I_G^*(A')$, for re-sort dataset $A$ by Eqn. (3) and for the corresponding encrypted dataset $A' = \{([\![a_{\langle 1 \rangle}]\!]_1, y_{\langle 1 \rangle}), \cdots, ([\![a_{\langle n \rangle}]\!]_1, y_{\langle n \rangle})\}$ from Eqns. (4)-(5).*

This theorem shows that our encryption could preserve the minimum Gini impurity over encrypted data. The detailed proof is presented in Appendix B, which involves the proof of piecewise monotonicity of $I_G(A, a)$ w.r.t. splitting point $a$, and then solves the minimum splitting point on plaintexts, as well as the corresponding point on encrypted data.

## 3.2 Binary Search Tree for Encryption

We now present new binary search tree to encrypt $a_1, \cdots, a_n$ dynamically, especially for un-ordered dataset $A = \{(a_1, y_1), \cdots, (a_n, y_n)\}$, or when example $(a_i, y_i)$ arrives in a streaming data. We begin with an alternative structure for binary search tree to maintain several samples on a node from Eqns. (4)-(5), rather than previous only one sample [88, 89]. Our new structure is given by

      **Struct Tree** {Plaintext *samples*; Ciphertext $cipher_1, cipher_2$; Tree *left*, *right*} .

The *samples* stores one or multiple samples from $A$, and $cipher_1$ and $cipher_2$ are the first and second ciphertext in Eqn. (5), and *left* and *right* denote left and right child of the current node, respectively.

We initialize an empty tree $\mathcal{BT} = \emptyset$ and set its $cipher_1 = c_{\max}/2$ with $c_{\max} = 2^{\lambda \log_2 n}$, and then we construct binary search tree iteratively. We maintain an interval $[t_{\min}, t_{\max}]$ in each iteration so as

---
**Algorithm 2** Splitting a node for encryption

---
**Input:** Example $(a_i, y_i)$, node $t$ of binary search tree $\mathcal{BT}$, and interval $[t_{\min}, t_{\max}]$
**Output:** Updated node $t$

   Initialize an empty node $l$ with $l.samples = \{(a_j, y_j) \in t.samples\colon a_j < a_i\}$
   **if** $l.samples \neq \emptyset$ **then**
     **if** $t.left \neq \emptyset$ **then**
       Set $l.cipher_1$ according to Eqn. (8), and update $l.left = t.left$, $t.left = l$
     **else**
       Set $l.cipher_1$ according to Eqn. (9), and update $t.left = l$
     **end if**
   **end if**
   Initialize an empty node $r$ with $r.samples = \{(a_j, y_j) \in t.samples\colon a_j > a_i\}$
   **if** $r.samples \neq \emptyset$ **then**
     **if** $t.right \neq \emptyset$ **then**
       Set $r.cipher_1$ according to Eqn. (10), and update $r.right = t.right$, $t.right = r$
     **else**
       Set $r.cipher_1$ according to Eqn. (11), and update $t.right = r$
     **end if**
   **end if**
   Update $t.samples = t.samples \setminus l.samples \setminus r.samples$

---

to keep the increasing order of ciphertexts $c_1, c_2, \cdots, c_s$ in Eqn. (5). During the $i$-th iteration, we receive a sample $(a_i, y_i)$, and then take two steps as follows:

**Step-I: Search a node for sample $(a_i, y_i)$ in binary search tree $\mathcal{BT}$**

Let $t$ be a node pointer with the initialization of the root of $\mathcal{BT}$. We search a path downward in $\mathcal{BT}$ by comparing with $a_i$, and the search will terminate when $t$ is a leaf node or an empty node.

For an internal node $t$, the search continues to its left child and updates $t_{\max} = t.cipher_1$ if

$$\text{the left child } t.left \neq \emptyset \quad \text{and} \quad a_i < \max\{a_j\colon (a_j, y_j) \in t.left.samples\} \,;$$

and the search continues to its right child and updates $t_{\min} = t.cipher_1$ if

$$\text{the right child } t.right \neq \emptyset \quad \text{and} \quad a_i > \min\{a_j\colon (a_j, y_j) \in t.right.samples\} \,;$$

otherwise, the search terminates. This procedure can be easily implemented with a while loop.

It is necessary to consider two special cases after the above search. We update $t = t.left$ if

$$t.left \neq \emptyset, \ a_i < \min\{a_j : (a_j, y_j) \in t.samples\} \text{ and } y_i = y_j \text{ for all } (a_j, y_j) \in t.left.samples \,. \quad (6)$$

In a similar manner, we update $t = t.right$ if

$$t.right \neq \emptyset, \ a_i > \max\{a_j : (a_j, y_j) \in t.samples\} \text{ and } y_i = y_j \text{ for all } (a_j, y_j) \in t.right.samples \,. \quad (7)$$

**Step-II: Update the binary search tree $\mathcal{BT}$**

After Step-I, we could find a node $t$ for sample $(a_i, y_i)$ and the corresponding interval $[t_{\min}, t_{\max}]$. We directly append the example $(a_i, y_i)$ into $t.samples$ if $y_i = y_j$ for every $(a_j, y_j) \in t.samples$; otherwise, it is necessary to split the node $t$ according to $a_i$.

We initialize an empty node $l$ with $l.samples = \{(a_j, y_j) \in t.samples\colon a_j < a_i\}$, and it is sufficient to consider $l.samples \neq \emptyset$. If $t.left \neq \emptyset$, then we set

$$l.cipher_1 = (t.left.cipher_1 + t.cipher_1)/2 + \xi \quad \text{s.t.} \quad t.left.cipher_1 < l.cipher_1 < t.cipher_1 \,, \quad (8)$$

and update $l.left = t.left$, $t.left = l$; otherwise, we set

$$l.cipher_1 = (t_{\min} + t.cipher_1)/2 + \xi \quad \text{s.t.} \quad l.cipher_1 \in (t_{\min}, t.cipher_1) \,, \quad (9)$$

and update $t.left = l$. Here, $\xi$ is a random number sampled from $\mathcal{N}(0, 1)$, and notice that we may randomly sample $\xi$ multiple times so that the condition holds in Eqns (8)-(9), respectively.

---

**Algorithm 3** Finding the best splitting feature and position

---

**Input:** Encrypted datasets $[\![S_n^t]\!]$, available splitting feature and position $[\![s]\!]_{i=1}^j$, and secret key $k_{\text{sec}}$
**Output:** index $i^*$
%% Server:
    **for** $i \in [j]$ **do**
        Calculate Gini impurity $I_G([\![S_n^t]\!], [\![s]\!]_i)$ from Eqn. (12) w.r.t splitting feature and position $[\![s]\!]_i$
    **end for**
    Send ciphertexts $\{I_G([\![S_n^t]\!], [\![s]\!]_i)\}_{i \in [j]}$ to the client
%% Client:
    Get the decrypted $\{\text{Dec}(k_{\text{sec}}, I_G([\![S_n^t]\!], [\![s]\!]_i))\}_{i \in [j]}$
    Set $i^* = -1$ if $\text{Dec}(k_{\text{sec}}, I_G([\![S_n^t]\!], [\![s]\!]_i)) = 0$ for every $i \in [j]$; otherwise, set $i^*$ by Eqn. (13)
    Send $i^*$ to the server

---

We make similar update for the right child of node $t$: initialize an empty node $r$ with $r.samples = \{(a_j, y_j) \in t.samples : a_j > a_i\}$, and consider $r.samples \neq \emptyset$. If $t.right \neq \emptyset$, then we set

$$r.cipher_1 = (t.cipher_1 + t.right.cipher_1)/2 + \xi \text{ s.t. } t.cipher_1 < r.cipher_1 < t.right.cipher_1 , \quad (10)$$

and update $r.right = t.right$, $t.right = r$; otherwise, we set

$$r.cipher_1 = (t.cipher_1 + t_{\max})/2 + \xi \quad \text{s.t.} \quad r.cipher_1 \in (t.cipher_1, t_{\max}) , \quad (11)$$

and update $t.right = r$. Algorithm 2 presents the detailed descriptions on the splitting of node $t$.

Algorithm 1 presents an overview of our Gini-impurity-preserving encryption, and the decryption is given in Appendix A. Our scheme does not only keep the minimum Gini impurity, but also change frequencies to prevent decryption from frequencies, which is also beneficial for encryption [90]. Our scheme takes an average of $O(n \log n)$ computational complexity, since it requires $O(\log n)$ and $O(1)$ computational complexities to search and update a node in each iteration, respectively. Finally, the average and worst space complexities are $O(\log n)$ and $O(n)$ for our encryption, respectively.

### 3.3 Security Analysis

For ciphertext vector $[\![a]\!] = ([\![a]\!]_1, [\![a]\!]_2)$ in Eqn. (5), it suffices to discuss the first ciphertext $[\![a]\!]_1$, since the security of $[\![a]\!]_2$ has been analyzed in homomorphic encryption CKKS [42]. Following semantic security against chosen plaintext attacks [89, 91], we define a security game $\text{Game}_{\text{GIPCPA}}$:

- An adversary chooses two sequences with distinct plaintexts $\{a_1^0, \cdots, a_n^0\}$ and $\{a_1^1, \cdots, a_n^1\}$, and sends them to a challenger;

- The challenger flips an unbiased coin $b \in \{0, 1\}$ to select $\{a_1^b, \cdots, a_n^b\}$, and randomly sets their corresponding labels $\{y_1^b, \cdots, y_n^b\}$ with each $y_i^b$ drawn independently and uniformly over $[\tau]$. The challenger encrypts $\{a_1^b, \cdots, a_n^b\}$ by Eqns. (4) and (5), and sends the ciphertexts to the adversary;

- The adversary outputs a guess of $b$, i.e., which sequence is selected for encryption.

We then introduce the security against Gini-impurity-preserving chosen plaintext attack as follows.

**Definition 2.** A scheme is said to be indistinguishable under Gini-impurity-preserving chosen plaintext attack if the probability of outputs with the correct guess $b$ is negligible for the adversary $\mathcal{A}$ in $\text{Game}_{\text{GIPCPA}}$, that is,

$$\Pr[\mathcal{A}(\text{Game}_{\text{GIPCPA}}) = b] < 1/2 + \text{ small constant} .$$

The following theorem shows that our encrypted plaintexts sequences are indistinguishable.

**Theorem 3.** *Our scheme for the first ciphertexts $[\![a_1]\!]_1, [\![a_2]\!]_1, \cdots, [\![a_n]\!]_1$ in Section 3.2 is security against Gini-impurity-preserving chosen plaintext attack.*

The detailed proof is presented in Appendix C, and the basic idea is inspired from [88]. We take induction on $n$ to show that data point $(a_{i+1}^b, y_{i+1})$ affects the constructed binary search trees with the same probability as $b = 0$ and $b = 1$, and then the ciphertexts of data points $(a_{i+1}^b, y_{i+1})$ also follow the same distribution, i.e.,

$$P\left([\![a_1^0]\!], \cdots, [\![a_{i+1}^0]\!] | a_1^0, \cdots, a_{i+1}^0\right) = P\left([\![a_1^1]\!], \cdots, [\![a_{i+1}^1]\!] | a_1^1, \cdots, a_{i+1}^1\right) .$$

**Table 2:** Datasets

| Datasets | #Inst | #Feat | Datasets | #Inst | #Feat | Datasets | #Inst | #Feat | Datasets | #Inst | #Feat |
|---|---|---|---|---|---|---|---|---|---|---|---|
| wdbc | 569 | 30 | adver | 3,279 | 1,558 | ailerons | 13,750 | 41 | adult | 48,842 | 14 |
| cancer | 569 | 31 | bibtex | 7,396 | 1,836 | house | 22,784 | 16 | mnist | 70,000 | 780 |
| breast | 699 | 9 | phpB0 | 7,797 | 617 | a9a | 32,563 | 123 | miniboone | 72,998 | 51 |
| diabetes | 768 | 8 | pendigits | 10,992 | 16 | amazon | 32,769 | 9 | runwalk | 88,588 | 6 |
| german | 1,000 | 24 | phish | 11,055 | 30 | bank | 45,211 | 17 | covtype | 581,012 | 54 |

## 4 Encrypted Random Forests

For encrypted random forests, we follow the popular client-server protocols [51, 65, 66, 88]. A client encrypts training and testing data, and transfers encrypted data to an honest-but-curious server. The server trains random forests from the encrypted data with the aid of client, and finally returns predictions on encrypted testing data.

### Encryption for training and testing datasets

Recall training data $S_n = \{(\boldsymbol{x}_1, y_1), \cdots, (\boldsymbol{x}_n, y_n)\}$ with $\boldsymbol{x}_i = (x_{i,1}, \cdots, x_{i,d})$. The client constructs $d$ binary search trees $\mathcal{BT}_1, \mathcal{BT}_2, \cdots, \mathcal{BT}_d$ according to Algorithm 1 over different dimensional features and labels in $S_n$, where $\mathcal{BT}_j$ is used to encrypt features $\{x_{1,j}, \cdots, x_{n,j}\}$ for $j \in [d]$.

We take the homomorphic encryption CKKS [42] to encrypt training labels $y_1, \cdots, y_n$. Each label $y_i$ is encoded with a vector of size $\tau$ by one-hot method, and we encrypt the vector by homomorphic encryption CKKS with a public key $k_{pub}$. The ciphertexts $[\![y_i]\!] = [[\![y_{i,1}]\!], \cdots, [\![y_{i,\tau}]\!]]$ is given by

$$[\![y_{i,j}]\!] = \begin{cases} \text{Enc}(k_{\text{pub}}, 1) & \text{for } j = y_i, \\ \text{Enc}(k_{\text{pub}}, 0) & \text{otherwise.} \end{cases}$$

We obtain the final training data $[\![S_n]\!] = \{([\![\boldsymbol{x}_1]\!], [\![y_1]\!]), \cdots, ([\![\boldsymbol{x}_n]\!], [\![y_n]\!])\}$.

Let $\tilde{S}_{n'} = \{\tilde{\boldsymbol{x}}_1, \cdots, \tilde{\boldsymbol{x}}_{n'}\}$ be a testing data with instance $\tilde{\boldsymbol{x}}_i = (\tilde{x}_{i,1}, \cdots, \tilde{x}_{i,d})$. For every plaintext $\tilde{x}_{i,j}$ with $i \in [n']$ and $j \in [d]$, we search a node $t$ in the binary search tree $\mathcal{BT}_j$, similarly to the node search (Step-I) in Section 3.2, and obtain its ciphertext $[\![\tilde{x}_{i,j}]\!] = [t.cipher, \text{Enc}(k_{\text{pub}}, i)]$. We have the encrypted testing data $[\![\tilde{S}_{n'}]\!] = \{[\![\tilde{\boldsymbol{x}}_1]\!], \cdots, [\![\tilde{\boldsymbol{x}}_{n'}]\!]\}$.

### Construction on encrypted random forests

Encrypted random forests consist of individual decision trees $\mathcal{DT}_1, \cdots, \mathcal{DT}_m$, where each tree $\mathcal{DT}_i$ is constructed as follows. We first take a bootstrap sample $[\![S'_n]\!]$ from $[\![S_n]\!]$, and initialize $\mathcal{DT}_i$ with one node of data $[\![S'_n]\!]$. We repeat the following procedure recursively for each leaf node, until the number of training samples is smaller than $\alpha$, or all instances have the same label in the leaf node:

- Select a $k$-subset B from $d$ available features randomly without replacement;
- Find the best splitting feature in B and position by Gini impurity from the encrypted data;
- Split the current node into left and right children via the best splitting position and feature.

Such construction is essentially similar to original random forests [1], whereas we require a different way to find the best splitting feature and position based on Gini impurity from the encrypted data.

Let $t$ be the current leaf node for further splitting with the encrypted training data $[\![S_n^t]\!] \subseteq [\![S_n]\!]$, and $[\![s]\!]_1, \cdots, [\![s]\!]_{\jmath}$ denote all possible splitting features and positions in the scope of the corresponding feature subset B from $[\![S_n^t]\!]$. Here, the information of feature and position can be derived from the corresponding index $i \in [\jmath]$ and subset B.

For each $i \in [\jmath]$, the server partitions the current encrypted training data $[\![S_n^t]\!]$ into left and right subsets, i.e., $[\![S_n^t]\!]_i^l$ and $[\![S_n^t]\!]_i^r$, according to the splitting feature and position $[\![s]\!]_i$. Let $n_l$ and $n_r$ be the number of training examples in $[\![S_n^t]\!]_i^l$ and $[\![S_n^t]\!]_i^r$, respectively, and denote by

$$[\![S_n^t]\!]_i^l = \{([\![\boldsymbol{x}_1^l]\!], [\![y_1^l]\!]), \cdots, ([\![\boldsymbol{x}_{n_l}^l]\!], [\![y_{n_l}^l]\!])\} \quad \text{and} \quad [\![S_n^t]\!]_i^r = \{([\![\boldsymbol{x}_1^r]\!], [\![y_1^r]\!]), \cdots, ([\![\boldsymbol{x}_{n_r}^r]\!], [\![y_{n_r}^r]\!])\} .$$

From Eqn. (1), we have Gini impurity

$$I_G([\![S_n^t]\!], [\![s]\!]_i) = \left[ \frac{n_l}{n_l + n_r} \otimes I_G([\![S_n^t]\!]_i^l) \right] \oplus \left[ \frac{n_r}{n_l + n_r} \otimes I_G([\![S_n^t]\!]_i^r) \right], \tag{12}$$

**Table 3:** Comparisons of prediction accuracies (mean±std). ●/○ indicates that our encrypted random forests are significantly better/worse than other compared random forests (pairwise $t$-tests at $95\%$ significance level). 'NA' means that no results were obtained after running out $10^6$ seconds (about 11.6 days).

| Dataset | Our encrypted RFs | Original RFs | AnonyRFs | DiffPrivRFs | PPD-ERTs | PivotRFs | MulPRFs | HEldpRFs |
|---|---|---|---|---|---|---|---|---|
| wdbc | .9525±.0141 | .9617±.0018 | .9091±.0205● | .8998±.0024● | .9222±.0037● | .9609±.0101 | .9510±.0114 | .9195±.0029● |
| cancer | .9766±.0082 | .9824±.0143 | .9271±.0016● | .9034±.0578● | .9600±.0022● | .9510±.0130● | .9656±.0102 | .9823±.0024 |
| breast | .9855±.0012 | .9881±.0011 | .9657±.0021● | .9271±.0515● | .9678±.0129● | .9806±.0086 | .9769±.0107 | .9275±.0023● |
| german | .7939±.0124 | .8033±.0205 | .7300±.0214● | .7400±.0141● | .7610±.0168● | .7533±.0122● | .7823±.0154 | .7043±.0027● |
| diabetes | .7641±.0093 | .7677±.0309 | .7193±.0023● | .7328±.0124● | .7448±.0193 | .7419±.0061● | .7611±.0035 | .7478±.0193● |
| adver | .9851±.0011 | .9888±.0014 | .9278±.0018● | .9390±.0051● | NA | .9664±.0043● | NA | NA |
| bibtex | .7907±.0054 | .7749±.0027● | .7425±.0009● | .7200±.0130● | NA | .7461±.0193● | NA | NA |
| phpB0 | .9380±.0024 | .9585±.0043○ | .8641±.0009● | .8920±.0031● | NA | NA | NA | NA |
| pendigits | .9917±.0024 | .9906±.0016 | .9072±.0104● | .9154±.0126● | .9639±.0048● | .9070±.0130● | NA | NA |
| phish | .9798±.0026 | .9716±.0018 | .9032±.0014● | .9318±.0089● | .9555±.0125● | .9454±.0067● | .9401±.0102● | NA |
| ailerons | .8795±.0027 | .8819±.0015 | .8104±.0105● | .8322±.0091● | .8589±.0043● | .8571±.0082● | .8766±.0025 | NA |
| house | .8794±.0007 | .8913±.0039○ | .8255±.0011● | .8475±.0025● | .8541±.0149● | .8508±.0016● | .8742±.0023 | NA |
| a9a | .8321±.0011 | .8303±.0012 | .8046±.0027● | .7909±.0084● | .8345±.0144 | .8314±.0071 | .8051±.0102● | NA |
| amazon | .9491±.0109 | .9478±.0060 | .9193±.0024● | .9104±.0035● | .9221±.0024● | .9401±.0128 | .9400±.0032 | NA |
| bank | .8992±.0118 | .9029±.0104 | .8499±.0089● | .8517±.0064● | .8940±.0147 | .8940±.0091 | .8827±.0108 | NA |
| adult | .8663±.0019 | .8691±.0018 | .8206±.0032● | .8355±.0053● | .8452±.0106● | .8243±.0076● | .8594±.0103 | NA |
| mnist | .9674±.0105 | .9763±.0101 | .9362±.0006● | .9059±.0157● | NA | NA | NA | NA |
| miniboone | .9497±.0018 | .9518±.0013 | .8977±.0101● | .9111±.0104● | .9301±.00021● | .9501±.0011 | NA | NA |
| runwalk | .9784±.0014 | .9798±.0032 | .9523±.0024● | .9401±.0040● | .9572±.0074● | .9511±.0071● | NA | NA |
| covtype | .9787±.0042 | .9650±.0104● | .9112±.0015● | .9407±.0018● | .9569±.0134● | NA | NA | NA |
| win/tie/loss | | 2/16/2 | 20/0/0 | 20/0/0 | 17/3/0 | 14/6/0 | 10/10/0 | 19/1/0 |

where $I_G(\llbracket S_n^t \rrbracket_i^l) = 1 \ominus p_l \odot p_l$ and $I_G(\llbracket S_n^t \rrbracket_i^r) = 1 \ominus p_r \odot p_r$, with

$$p_l = (1/n_l) \otimes (\llbracket y_1^l \rrbracket \oplus, \cdots, \oplus \llbracket y_{n_l}^l \rrbracket) \ \text{ and } \ p_r = (1/n_r) \otimes (\llbracket y_1^r \rrbracket \oplus, \cdots, \oplus \llbracket y_{n_r}^r \rrbracket) \ .$$

Here, $\otimes$, $\odot$, $\oplus$ and $\ominus$ denote the CKKS element-wise homomorphic multiplication, dot, addition and subtraction functions, respectively, as in the work of [42].

The client gets plaintexts $\{\text{Dec}(k_{\text{sec}}, I_G(\llbracket S_n^t \rrbracket, \llbracket s \rrbracket_i))\}_{i=1}^j$ by decrypting with the secret key $k_{\text{sec}}$, when the server sends ciphertexts $\{I_G(\llbracket S_n^t \rrbracket, \llbracket s \rrbracket_i)\}_{i=1}^j$. If all instances have the same label in $\llbracket S_n^t \rrbracket$, then we have $\text{Dec}(k_{\text{sec}}, I_G(\llbracket S_n^t \rrbracket, \llbracket s \rrbracket_i)) = 0$ for each $i \in [j]$, and we set $i^* = -1$; otherwise, we set $i^*$ as

$$i^* \in \arg\min_{i \in [j]} \left\{ \text{Dec}(k_{\text{sec}}, I_G(\llbracket S_n^t \rrbracket, \llbracket s \rrbracket_i)) \right\} . \tag{13}$$

The client sends index $i^*$ to the server for further splitting. Algorithm 3 presents the detailed descriptions on finding the best splitting feature and position.

For encrypted decision tree, the client requires the $O(\kappa)$ computational complexity with $\kappa$ leaves nodes, since the client performs constant basic operations for each node. The server takes the $O(\kappa \bar{j} \tau n)$ computational complexity for Eqn. (12), where $\bar{j}$ is an average of number of possible splitting features and positions, and $\tau$ and $n$ are the number of labels and training examples, respectively.

Our method takes $O(h)$ communication rounds of $O(\kappa \bar{j})$ communication bandwidth to train an encrypted decision tree of height $h$. This is because we consider the breadth-first search and aggregate all nodes in the same height and send to the client with a single message at one time.

We do not require bootstrapping for homomorphic encryption in 3-depth homomorphic multiplicative, since we independently compute the splitting feature and position for each node from Eqn. (12). This is different from previous encrypted decision trees [51, 62], which could take expensive computational complexity for bootstrapping [40, 92].

**Prediction on encrypted testing dataset**

After getting decision trees $\mathcal{DT}_1, \cdots, \mathcal{DT}_m$, we predict label $\llbracket \tilde{y}_i \rrbracket = \mathcal{DT}_1(\llbracket \tilde{\boldsymbol{x}}_i \rrbracket) \oplus \cdots \oplus \mathcal{DT}_m(\llbracket \tilde{\boldsymbol{x}}_i \rrbracket)$ for test instance $\llbracket \tilde{\boldsymbol{x}}_i \rrbracket \in \llbracket \tilde{S}_{n'} \rrbracket$. The server sends ciphertexts $\{\llbracket \tilde{y}_1 \rrbracket, \cdots, \llbracket \tilde{y}_{n'} \rrbracket\}$ to the client, and the client decrypts those ciphertexts, and gets the final plaintext label by $\tilde{y}_i = \arg\max_{j \in [\tau]} \{\text{Dec}\llbracket \tilde{y}_{i,j} \rrbracket\}$.

During such prediction process, the server requires the $O(h)$ computational complexity, since we search from the root to leaf node of tree. The client takes $O(1)$ rounds of communication and communication bandwidth to transfer the testing data and predicting ciphertext without interaction.

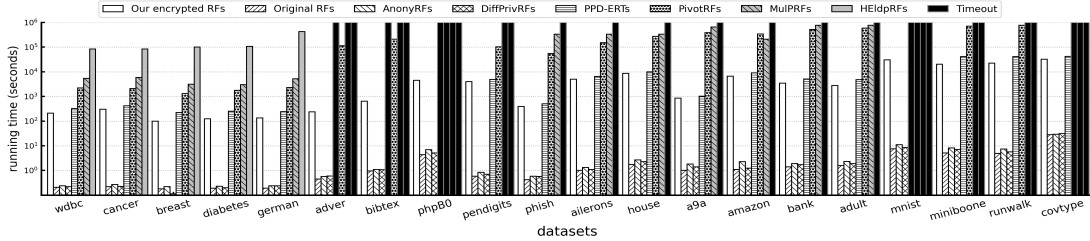

**Figure 2:** Comparisons of training running time on different random forests. Notice that the y-axis is in log-scale, and full black columns imply that no result was obtained after running out $10^6$ seconds (about 11.6 days).

## 5 Experiment

We conduct experiments on 20 datasets[2] as summarized in Table 2. Most datasets have been well-studied in previous random forests. In addition to the original (plaintexts) random forests [1], we compare with six state-of-the-art privacy-preserving random forests in recent years.

- AnonyRFs: random forests based on anonymization with a top-down greedy search [59];
- DiffPrivRFs: random forests based on differential privacy [93];
- PPD-ERTs: extremely randomized trees from distributed structured data [64];
- PivotRFs: random forests based on a hybrid of threshold partially homomorphic encryption and secure multiparty computation techniques [62];
- MulPRFs: random forests based on the secure multiparty computation [94];
- HEldpRFs: random forests with fully homomorphic encryption and low-degree polynomial approximations [51].

For all random forests, we train 100 individual decision trees, and randomly select $\lfloor\sqrt{d}\rfloor$ candidate features during node splitting. We set $\alpha = 10$ for datasets of size smaller than 20,000 for our encrypted random forests; otherwise, set $\alpha = 100$, following [95]. For multi-class datasets, we take the one-vs-all method for MulPRFs, since it is limited to binary classification. Other parameters are set according to their respective references, and more details can be found in Appendix D.

**Experimental comparisons**

The performance is evaluated by five trials of 5-fold cross validation, and final prediction accuracies are obtained by averaging over these 25 runs, as summarized in Table 3. It is evident that our encrypted random forests take comparable performance with original random forests [1] on plaintexts, which nicely supports our Theorem 1 on the preservation of minimum Gini impurity in the construction of random forests. Our encrypted random forests are also comparable to MulPRFs if they can obtain results within $10^6$ seconds (about 11.6 days), since MulPRFs are essentially similar to original random forests, yet with different implementation of secure multi-party computation.

As can be seen from Table 3, our random forests take significantly better performance than AnonyRFs and DiffPrivRFs, since the win/tie/loss counts show that our random forests win for most times and never lose. This is because AnonyRFs combine features by anonymization, while DiffPrivRFs add perturbations to features via differential privacy, therefore, both of them cause information lost in privacy process. Our random forests also achieve better performance than PivotRFs, since PivotRFs have to limit trees' depth for random forests due to heavy computations for HE and communications for secure multi-party computation.

Our random forests also outperform PPD-ERTs and HEldpRFs if results are obtained in $10^6$ seconds, since PPD-ERTs adopt completely-random splitting, rather than selecting the minimum Gini impurity, while HEldpRFs take homomorphic encryption on features and employ low-degree polynomial approximation. Those approaches have modified the structures of original random forests.

---

[2]Downloaded from *www.openml.org*

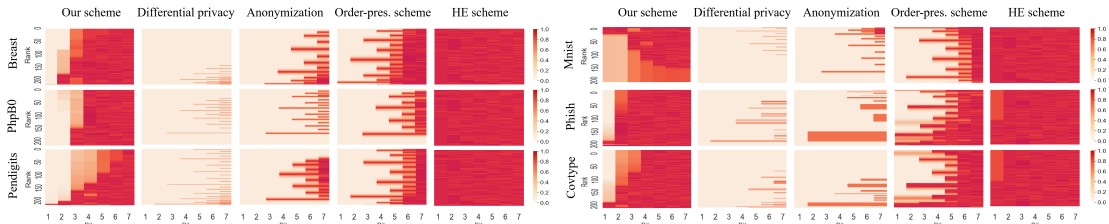

**Figure 3:** Security comparisons for different schemes: the more red the area, the higher the security.

**Running time**

All experiments are performed by c++ on the Ubuntu with 256GB main memory (AMD Ryzen Threadripper 3970X). We compare the training running time of our encrypted random forests and others, and the average CPU time (in seconds) is shown in Figure 2.

As expected, original random forests take the least running time over raw datasets without privacy preservation. Our encrypted random forests take larger running time than AnonyRFs and DiffPrivRFs because they are essentially similar to original random forests, yet with some simple modifications or perturbations on features. Our encrypted random forests take better performance and higher security.

Our encrypted random forests take smaller running time than PPD-ERTs, PivotRFs, MulPRFs and HEldpRFs, in particular for large datasets or high-dimensional datasets, where no results are obtained even after running out $10^6$ seconds (almost 11.6 days). Because PPD-ERTs, PivotRFs and MulPRFs require expensive communication cost for multi-parity computation, while PivotRFs and HEldpRFs take heavy computation costs on HE scheme.

**Security analysis**

We present security analysis for the first ciphertext $[\![a]\!]_1$ in ciphertext vector $[\![a]\!] = ([\![a]\!]_1, [\![a]\!]_2)$, and the second ciphertext $[\![a]\!]_2$ can be ensured by HE scheme. We compare with four state-of-the-art encryptions: differential privacy [93], anonymization [59], order-preserving scheme [96] and HE scheme [42]. Here, we present results of six datasets and randomly selecting one feature, and trends are similar on other dimensions and datasets. More results can be found in Appendix D.

Figure 3 shows the comparison results, and we take the bitwise leakage matrices to measure the security as in [97]: the more red the area, the higher the security. As expected, HE scheme presents the highest security, yet with heavy computational costs, for example, no results are obtained for datasets of size exceeding 3000 even after running out $10^6$ seconds. It is also observed that our scheme presents higher security than the other three schemes, since those schemes simply present perturbations, compression or preserve the entire order information regardless of learning ingredients. In comparison, our scheme could make a good balance between security and computational cost.

## 6   Conclusion

This work takes one step on data encryption from some crucial ingredients of learning algorithm. We present a new encryption to preserve data's Gini impurity, which plays a crucial role during the construction of random forests. For random forests, we encrypt data features based on our Gini-impurity-preserving scheme, and take the homomorphic encryption scheme CKKS to encrypt data labels. Both theoretically and empirically, we validate the effectiveness, efficiency and security of our proposed method. An interesting work is to exploit other learning ingredients, such as gini index and information gain, for data encryption in the future.

## Acknowledgements

The authors want to thank the reviewers for their helpful comments and suggestions. This research was supported by National Key R&D Program of China (2021ZD0112802), NSFC (61921006, 62376119), CAAI-Huawei MindSpore Open Fund, and Fundamental Research Funds for the Central Universities (2023300246). W. Gao is the corresponding author of this paper.

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

**Algorithm 4** Decryption

---

**Input:** Tree node $t$ of $\mathcal{BT}$, ciphertext $[\![a_i]\!]$
**Output:** plaintext $a_i$
    **while** $[\![a_i]\!]_1 \neq t.cipher_1$ **do**
      **if** $[\![a_i]\!]_1 > t.cipher_1$ **then**
        $t = t.right$
      **else if** $[\![a_i]\!]_1 < t.cipher_1$ **then**
        $t = t.left$
      **end if**
    **end while**
    Return $a_i = t.samples[\text{Dec}(k_{\text{sec}}, [\![a_i]\!]_2)]$

---

# A   Detailed Decryption for Our Encryption Method

## A.1   Decryption for Our Encryption in Section 3.1

We present the decryption for ciphertext $[\![a_i]\!] = ([\![a_i]\!]_1, [\![a_i]\!]_2)$ in Eqn. (5) by the following steps:

- Find the partition $\mathcal{I}_j$ according to $[\![a_i]\!]_1$;
- Decrypt ciphertext $[\![a_i]\!]_2$ by the CKKS secret key $k_{\text{sec}}$, and get index $\tau = \text{Dec}(k_{\text{sec}}, [\![a_i]\!]_2)$ in partition $\mathcal{I}_j$.
- Obtain the plaintext $a_i$ as the $\tau$-th sample in partition $\mathcal{I}_j$.

## A.2   Decryption for Our Encryption of Binary Search Tree in Section 3.2

We decrypt a ciphertext $[\![a_i]\!] = ([\![a_i]\!]_1, [\![a_i]\!]_2)$ based on binary search tree $\mathcal{BT}$ (in Section 3.2) and the CKKS secret key $k_{\text{sec}}$ by the following two steps, and Algorithm 4 presents the details of decryption:

- Let $t$ be a node pointer with the initialization of the root of binary search tree $\mathcal{BT}$. We then search a path downward in $\mathcal{BT}$ by comparing with $[\![a_i]\!]_1$. The search continues to its left child if $[\![a_i]\!]_1 < t.cipher_1$ and update $t = t.left$; the search continues to its right child if $[\![a_i]\!]_1 > t.cipher_1$ and update $t = t.right$ until $[\![a_i]\!]_1 = t.cipher_1$.
- Decrypt ciphertext $[\![a_i]\!]_2$ by the CKKS secret key $k_{\text{sec}}$, and get index $\tau = \text{Dec}(k_{\text{sec}}, [\![a_i]\!]_2)$ in $t.samples$. Then we use the index $\tau$ to get the plaintext $a_i = t.samples[\tau]$.

## A.3   Formal Definition of Our Gini-impurity-preserving Encryption

We present a formal definition of our Gini-impurity preserving encryption as follows:

- $S \leftarrow \text{KeyGen}(t_{\max})$: Generate the secret state $S$ by initializing binary search tree $\mathcal{BT} = \emptyset$, and a security parameter $c_{\max}$, which is a random number with $c_{\max} > n$. We maintain an interval $[t_{\min}, t_{\max}]$ in each secret state $S$ with $t_{\min} = 0$ and $t_{\max} = c_{\max}$ in the initial stage, so as to keep the order of ciphertexts $c_1, c_2, \cdots, c_s$ in Eqn. (5). In this way, the ciphertexts are random numbers with semi-order of plaintexts, and we have different ciphertext even for the same plaintexts.

- $S', [\![a_i]\!] \leftarrow \text{Encrypt}(S, a_i)$: Encrypt $a_i$ and update the secret state to $S'$ as for receiving a sample $(a_i, y_i)$ as follows:
  - Search a node for sample $(a_i, y_i)$ in binary search tree $\mathcal{BT}$ as shown in Algorithm 1. Let $t$ be a node pointer with the initialization of the root of $\mathcal{BT}$. We search a path downward in $\mathcal{BT}$ by comparing with $a_i$. The search will terminate when $t$ is a leaf or an empty node.
  - Update the binary search tree $\mathcal{BT}$. We directly append the example $(a_i, y_i)$ into $t.samples$ if $y_i = y_j$ for every $(a_j, y_j) \in t.samples$; otherwise, it is necessary to split the node $t$ according to $a_i$. Algorithm 2 presents the detailed descriptions on the splitting of node $t$.
  - Compute ciphertext $[\![a_i]\!]$ and update the state from $S$ to $S'$. Append example $(a_i, y_i)$ into $t.samples$ and update $t.cipher_2 = \text{Enc}(k_{\text{pub}}, |t.samples|)$. We then compute the ciphertext $[\![a_i]\!] = (t.cipher_1, t.cipher_2)$, and update the state from $S$ to $S'$ through our $\mathcal{BT}$.

- $a_i \leftarrow \text{Decrypt}(S', [\![a_i]\!])$: Solve plaintext $a_i$ for ciphertext $[\![a_i]\!]$ based on state $S'$ with binary search tree $\mathcal{BT}$ and the CKKS secret key $k_{\text{sec}}$ as follows:

  - Let $t$ be a node pointer with initialing the root of binary search tree $\mathcal{BT}$. We then search a path downward in binary search tree $\mathcal{BT}$ by comparing with $[\![a_i]\!]_1$. The search continues to its left child if $[\![a_i]\!]_1 < t.cipher_1$ and update $t = t.left$; the search continues to its right child if $[\![a_i]\!]_1 > t.cipher_1$ and update $t = t.right$ until $[\![a_i]\!]_1 = t.cipher_1$.

  - Decrypt ciphertext $[\![a_i]\!]_2$ by CKKS secret key $k_{\text{sec}}$, and get index $\tau = \text{Dec}(k_{\text{sec}}, [\![a_i]\!]_2)$ in $t.samples$. Then we use the index $\tau$ to get the plaintext $a_i = t.samples[\tau]$.

# B   Proof of Theorem 1

**Lemma 4.** *For dataset* $A = \{(a_1, y_1), \cdots, (a_n, y_n)\}$, *let* $\mathcal{I}_1, \mathcal{I}_2, \cdots, \mathcal{I}_s$ *be the corresponding partitions as defined by Eqn.* (4). *There exists a splitting point* $a^*$ *such that* $I_G(A, a^*) = I_G^*(A)$ *and*

$$a^* \in \bigcup_{i \in [s-1]} \{\max\{a_k \colon (a_k, y_k) \in \mathcal{I}_i\}/2 + \min\{a_k \colon (a_k, y_k) \in \mathcal{I}_{i+1}\}/2\} \ ,$$

*where* $I_G(A, a^*)$ *and* $I_G^*(A)$ *are defined by Eqns.* (1) *and* (2), *respectively.*

*Proof.* Without loss of generality, we assume that $a_1, a_2, \cdots, a_n$ are distinct elements. Our goal is to solve the optimal splitting point $a^* \in \arg\min_{a \in \mathbb{R}}\{I_G(A, a)\}$, and we begin with some notations used in our proof. For every label $j \in [\tau]$, we denote by

$$\nu_j = |\{i \in [n]\colon y_i = j\}| \ ,$$

i.e., the number of the label $j$ in dataset $A$. Let $a$ be a splitting point, which splits $A$ into left and right datasets $A_a^l$ and $A_a^r$, that is,

$$
\begin{aligned}
A_a^l &= \{(a_i, y_i)\colon a_i \leq a, (a_i, y_i) \in A\} \ , \\
A_a^r &= \{(a_i, y_i)\colon a_i > a, (a_i, y_i) \in A\} \ .
\end{aligned}
$$

For any given $a \in \mathbb{R}$ and $j \in [\tau]$, we further denote by

$$\nu_j^l = |\{i \in [n]\colon y_i = j, a_i \leq a\}| \ ,$$

i.e., the number of label $j$ in subsets $A_a^l$. This follows that

$$I_G(A, a) = w_l - w_l \sum_{j \in [\tau]} \frac{(\nu_j^l)^2}{|A_a^l|^2} + w_r - w_r \sum_{j \in [\tau]} \frac{(\nu_j - \nu_j^l)^2}{(n - |A_a^l|)^2} \ ,$$

where $w_l = |A_a^l|/n$, and $w_r = 1 - w_l$. In the following, we will explore the monotonicity of function $I_G(A, a)$ when

$$
\begin{aligned}
a &\geq \max\{a_k : (a_k, y_k) \in \mathcal{I}_{i-1}\}/2 + \min\{a_k : (a_k, y_k) \in \mathcal{I}_i\}/2 \\
a &\leq \max\{a_k : (a_k, y_k) \in \mathcal{I}_i\}/2 + \min\{a_k : (a_k, y_k) \in \mathcal{I}_{i+1}\}/2 \ ,
\end{aligned}
$$

for $i = 2, 3, \cdots, s - 1$. It is easy to observe that $\nu_j$ and $\nu_j^l$ keep constants except for $\nu_{j_*}^l$, where $j_*$ denotes the label of instances in $\mathcal{I}_i$. It remains to discuss the variable $\nu_{j_*}^l$, and we have

$$n^2 \frac{\partial I_G(A,a)}{\partial \nu_{j*}^l} = \frac{1}{n} \sum_{j\in[\tau]} \frac{(\nu_j^l)^2}{(w_l)^2} - 2\frac{\nu_{j*}^l}{w_l} - \frac{1}{n} \sum_{j\in[\tau]} \frac{(\nu_j - \nu_j^l)^2}{(w_r)^2} + 2\frac{(\nu_{j*} - \nu_{j*}^l)}{w_r}$$

$$= \frac{1}{n} \sum_{j\in[\tau]} \left( \left(\frac{\nu_j^l}{w_l}\right)^2 - \left(\frac{\nu_j - \nu_j^l}{w_r}\right)^2 \right) + 2\left(\frac{\nu_{j*} - \nu_{j*}^l}{w_r} - \frac{\nu_{j*}^l}{w_l}\right)$$

$$= \frac{1}{n} \sum_{j\in[\tau], j\neq j*} \left( \left(\frac{\nu_j^l}{w_l}\right)^2 - \left(\frac{\nu_j - \nu_j^l}{w_r}\right)^2 \right)$$

$$+ \frac{1}{n} \left( \left(\frac{\nu_{j*}^l}{w_l}\right)^2 - \left(\frac{\nu_{j*} - \nu_{j*}^l}{w_r)}\right)^2 \right) + 2\left(\frac{\nu_{j*} - \nu_{j*}^l}{w_r} - \frac{\nu_{j*}^l}{w_l}\right)$$

$$= \frac{1}{n} \sum_{j\in[\tau], j\neq j*} \left( \frac{(\nu_j^l)^2}{(w_l)^2} - \frac{(\nu_j - \nu_j^l)^2}{(w_r)^2} \right) + \left(\frac{\nu_{j*} - \nu_{j*}^l}{w_r} - \frac{\nu_{j*}^l}{w_l}\right) \left( 2 - \frac{\nu_{j*} - \nu_{j*}^l}{nw_r} - \frac{\nu_{j*}^l}{nw_l} \right) .$$

It is easy to observe that

$$0 \leq \frac{\nu_j - \nu_j^l}{w_r} \leq n \quad \text{and} \quad 0 \leq \frac{\nu_j^l}{w_l} \leq n \ \text{ for each } \ j \in [\tau] . \tag{14}$$

It is sufficient to consider two cases as follows:

- We consider the first case

$$\sum_{j\in[\tau], j\neq j*} \left( \left(\frac{\nu_j^l}{w_l}\right)^2 - \left(\frac{\nu_j - \nu_j^l}{w_r}\right)^2 \right) \geq 0 ,$$

and this follows that

$$\begin{aligned}
0 &\leq \sum_{j\in[\tau], j\neq j*} \left( \left(\frac{\nu_j^l}{w_l}\right)^2 - \left(\frac{\nu_j - \nu_j^l}{w_r}\right)^2 \right) \\
&= \sum_{j\in[\tau], j\neq j*} \left( \frac{\nu_j^l}{w_l} + \frac{\nu_j - \nu_j^l}{w_r} \right) \left( \frac{\nu_j^l}{w_l} - \frac{\nu_j - \nu_j^l}{w_r} \right) \\
&\leq \sum_{j\in[\tau], j\neq j*} 2n \left( \frac{\nu_j^l}{w_l} - \frac{\nu_j - \nu_j^l}{w_r} \right) = 2n \sum_{j\in[\tau], j\neq j*} \left( \frac{\nu_j^l}{w_l} - \frac{\nu_j - \nu_j^l}{w_r} \right) .
\end{aligned}$$

We have

$$n - \sum_{j\in[\tau], j\neq j*} \frac{\nu_j - \nu_j^l}{w_r} \geq n - \sum_{j\in[\tau], j\neq j*} \frac{\nu_j^l}{w_l} , \tag{15}$$

and it holds that

$$\frac{\nu_{j*} - \nu_{j*}^l}{w_r} \geq \frac{\nu_{j*}^l}{w_l} . \tag{16}$$

Combining with Eqns. (14)-(16), we have

$$\frac{\partial I_G(A,a)}{\partial \nu_{j*}^l} \geq 0 ,$$

which proves the increasing function of $I_G(A,a)$.

- We now consider the second case

$$\sum_{j\in[\tau],j\neq j_*} \left( \left( \frac{\nu_j^l}{w_l} \right)^2 - \left( \frac{\nu_j - \nu_j^l}{w_r} \right)^2 \right) < 0 \,,$$

and this follows that

$$\sum_{j\in[\tau]} \left( \left( \frac{\nu_j^l}{w_l} \right)^2 - \left( \frac{\nu_j - \nu_j^l}{w_r} \right)^2 \right) < \left( \frac{\nu_{j_*}^l}{w_l} \right)^2 - \left( \frac{\nu_{j_*} - \nu_{j_*}^l}{w_r} \right)^2$$

$$= \left( \frac{\nu_{j_*}^l}{w_l} + \frac{\nu_{j_*} - \nu_{j_*}^l}{w_r} \right) \left( \frac{\nu_{j_*}^l}{w_l} - \frac{\nu_{j_*} - \nu_{j_*}^l}{w_r} \right) < 2n \left( \frac{\nu_{j_*}^l}{w_l} - \frac{\nu_{j_*} - \nu_{j_*}^l}{w_r} \right) \,.$$

We have

$$n^2 \frac{\partial I_G(A,a)}{\partial \nu_{j_*}^l} = \frac{1}{n} \sum_{j\in[\tau]} \left( \left( \frac{\nu_j^l}{w_l} \right)^2 - \left( \frac{\nu_j - \nu_j^l}{w_r)} \right)^2 \right) + 2 \left( \frac{\nu_{j_*} - \nu_{j_*}^l}{w_r} - \frac{\nu_{j_*}^l}{w_l} \right)$$

$$< 2 \left( \frac{\nu_{j_*}^l}{w_l} - \frac{\nu_{j_*} - \nu_{j_*}^l}{w_r} \right) + 2 \left( \frac{\nu_{j_*} - \nu_{j_*}^l}{w_r} - \frac{\nu_{j_*}^l}{w_l} \right) = 0 \,,$$

which proves the decreasing function of $I_G(A,a)$.

In a summary, we prove the piecewise monotonicity of $I_G(A,a)$ for

$$a \geq \max\{a_k \colon (a_k,y_k) \in \mathcal{I}_{i-1}\}/2 + \min\{a_k \colon (a_k,y_k) \in \mathcal{I}_i\}/2$$
$$a \leq \max\{a_k \colon (a_k,y_k) \in \mathcal{I}_i\}/2 + \min\{a_k \colon (a_k,y_k) \in \mathcal{I}_{i+1}\}/2 \,,$$

with $i = 2,3,\cdots,s-1$. Moreover, it is easy to observe the monotonicity of $I_G(A,a)$ from $\nu_j^l = 0 (j \neq j_*)$ when

$$a \in (-\infty, (\max\{a_k : (a_k,y_k) \in \mathcal{I}_1\} + \min\{a_k : (a_k,y_k) \in \mathcal{I}_2\})/2] \;;$$

and from $\nu_j - \nu_j^l = 0 \ (j \neq j_*)$ when

$$a \in [(\max\{a_k : (a_k,y_k) \in \mathcal{I}_{s-1}\} + \min\{a_k : (a_k,y_k) \in \mathcal{I}_s\})/2, +\infty) \,.$$

It is not necessary to consider the splitting point $a^* > \max\{a_k : (a_k,y_k) \in \mathcal{I}_s\}$ with $|A_a^r| = 0$, as well as the splitting point $a^* < \min\{a_k : (a_k,y_k) \in \mathcal{I}_1\}$ with $|A_a^l| = 0$, i.e., without splitting dataset $A$. This completes the proof. □

**Proof of Theorem 1**

According to Lemma 6, we could find an optimal splitting point $a^*$ such that

$$a^* \in \bigcup_{i\in[s-1]} \left\{ \frac{\max\{a_k : (a_k,y_k) \in \mathcal{I}_i\} + \min\{a_k : (a_k,y_k) \in \mathcal{I}_{i+1}\}}{2} \right\} \,.$$

It is easy to observe that, for $i \in [s-1]$

$$I_G(A, (\max\{a_k : (a_k,y_k) \in \mathcal{I}_i\} + \min\{a_k : (a_k,y_k) \in \mathcal{I}_{i+1}\}/\,2) = I_G(A, (c_i + c_{i+1})/2) \,,$$

where $c_i$ is the identical ciphertext for those elements in $\in \mathcal{I}_i$, and we complete the proof. □

Based on Theorem 1, our encryption with binary search trees (Algorithm 1) can also preserve the minimum Gini impurity over encrypted data, which can be shown by the following theorem:

**Theorem 5.** *We have $I_G^*(A) = I_G^*(\hat{A})$, for re-sort dataset $A$ by Eqn. (3) and for the corresponding encrypted dataset $\hat{A} = \{([\![a_{\langle 1 \rangle}]\!]_1, y_{\langle 1 \rangle}), \cdots, ([\![a_{\langle n \rangle}]\!]_1, y_{\langle n \rangle})\}$ from Algorithm 1.*

*Proof.* Our constructed binary search tree $\mathcal{BT}$ (Algorithm 1) maintains several samples on a node. For each node $t$, we have $t.cipher_1 < t.right.cipher_1$ and $t.cipher_1 > t.left.cipher_1$. In this way, we can obtain a monotone increasing sequence $\mathcal{I}_1, \mathcal{I}_2, \cdots, \mathcal{I}_s$ by inorder traversing the built Tree $\mathcal{BT}$ in Algorithm 1. Each $\mathcal{I}_i$ for $j \in [s]$ contains several samples as follows:

$$
\begin{aligned}
\mathcal{I}_1 &= \left\{ (a_{\langle 1 \rangle}, y_{\langle 1 \rangle}), \cdots, (a_{\langle k_1 \rangle}, y_{\langle k_1 \rangle}) \right\} \\
\mathcal{I}_2 &= \left\{ (a_{\langle k_1+1 \rangle}, y_{\langle k_1+1 \rangle}), , \cdots, (a_{\langle k_2 \rangle}, y_{\langle k_2 \rangle}) \right\} \\
&\cdots \\
\mathcal{I}_s &= \left\{ (a_{\langle k_{s-1}+1 \rangle}, y_{\langle k_{s-1}+1 \rangle}), \cdots, (a_{\langle n \rangle}, y_{\langle n \rangle}) \right\},
\end{aligned}
\tag{17}
$$

where $a_{\langle i' \rangle} < a_{\langle j' \rangle}$ for $(a_{\langle i' \rangle}, y_{\langle i' \rangle}) \in \mathcal{I}_i, (a_{\langle j' \rangle}, y_{\langle j' \rangle}) \in \mathcal{I}_j$ and $i < j$.

For each $\mathcal{I}_i$, if there is only one identical label, i.e., $y_{\langle i \rangle} = y_{\langle i' \rangle}$ for every $(a_{\langle i \rangle}, y_{\langle i \rangle}), (a_{\langle i' \rangle}, y_{\langle i' \rangle}) \in \mathcal{I}_j$, then we have $I_G^*(A) = I_G^*(\hat{A})$ from Theorem 1. On the other hand, if the values are the same for all samples in $\mathcal{I}_j$ ($j \in [s]$), i.e., $a_{\langle i \rangle} = a_{\langle i' \rangle}$ for every $(a_{\langle i \rangle}, y_{\langle i \rangle}), (a_{\langle i' \rangle}, y_{\langle i' \rangle}) \in \mathcal{I}_j$, then this splitting value is preserved without changing the minimum Gini-impurity of random forests. Hence, we also have $I_G^*(A) = I_G^*(\hat{A})$, and this completes the proof. $\qquad\square$

## C  Proof of Theorem 3

Given two sequences of distinct plaintext $A^0 = \{a_1^0, a_2^0, \cdots, a_n^0\}$ and $A^1 = \{a_1^1, a_2^1, \cdots, a_n^1\}$, their corresponding labels are randomly set as follows:

- Sort $A^b = \{a_{\langle 1 \rangle}^b, a_{\langle 2 \rangle}^b, \cdots, a_{\langle n \rangle}^b\}$ in ascending order, i.e., $a_{\langle 1 \rangle}^b < a_{\langle 2 \rangle}^b < \cdots < a_{\langle n \rangle}^b$ for $b \in \{0, 1\}$.
- Set the corresponding labels $\{y_{\langle 1 \rangle}, y_{\langle 2 \rangle}, \cdots, y_{\langle n \rangle}\}$ randomly and independently from a uniform distribution on $[\tau]$.

Then, we have

**Lemma 6.** *For $a_1^b < a_2^b < \cdots < a_n^b$ with $b = \{0, 1\}$, we have the same Gini impurity for two sequences $A^0 = \{(a_1^0, y_1), (a_2^0, y_2), \cdots, (a_n^0, y_n)\}$ and $A^1 = \{(a_1^1, y_1), (a_2^1, y_2), \cdots, (a_n^1, y_n)\}$.*

*Proof.* Let $a_i^b$ be a splitting point for $b \in \{0, 1\}$ and $i \in [n]$, and we split $A^b$ into left and right datasets $A_{a_i^b}^{l,b}$ and $A_{a_i^b}^{r,b}$ according to $a_i^b$, that is,

$$ A_{a_i^b}^{l,b} = \{(a_1^b, y_1), (a_2^b, y_2), \cdots, (a_i^b, y_i)\} \text{ and } A_{a_i^b}^{r,b} = \{(a_{i+1}^b, y_{i+1}), (a_{i+2}^b, y_{i+2}), \cdots, (a_n^b, y_n)\}. $$

For $j \in [\tau]$, denote by $\nu_j^{l,b}$ and $\nu_j^{r,b}$ the cardinalities of subsets $A_{a_i^b}^{l,b}$ and $A_{a_i^b}^{r,b}$ with label $j$, respectively. Then, the Gini impurity of dataset $A^b$ and splitting point $a_i^b$ is given by

$$ I_G(A^b, a_i^b) = \frac{i}{n} - \frac{i}{n} \sum_{j \in [\tau]} \frac{(\nu_j^{l,b})^2}{(i)^2} + \frac{n-i}{n} - \frac{n-i}{n} \sum_{j \in [\tau]} \frac{(\nu_j^{r,b})^2}{(n-i)^2}. $$

For $b \in \{0, 1\}$, $A_{a_i^0}^{l,0}$ and $A_{a_i^1}^{l,1}$ have the same labels $\{y_1, y_2, \cdots, y_i\}$, and we have

$$ \sum_{j \in [\tau]} (\nu_j^{l,0})^2 = \sum_{j \in [\tau]} (\nu_j^{l,1})^2. $$

Similarly, we have $\sum_{j \in [\tau]} (\nu_j^{r,0})^2 = \sum_{j \in [\tau]} (\nu_j^{r,1})^2$, and this completes the proof. $\qquad\square$

We can show that adversary can not distinguish the ciphertext of $\{(a_{\langle 1 \rangle}^0, y_{\langle 1 \rangle}), \cdots, (a_{\langle n \rangle}^0, y_{\langle n \rangle})\}$ from that of $\{(a_{\langle 1 \rangle}^1, y_{\langle 1 \rangle}), \cdots, (a_{\langle n \rangle}^1, y_{\langle n \rangle})\}$ in a probabilistic perspective, i.e.,

$$
\begin{aligned}
&\Pr\left( [\![a_{\langle 1 \rangle}^0]\!], \cdots, [\![a_{\langle n \rangle}^0]\!] | (a_{\langle 1 \rangle}^0, y_{\langle 1 \rangle}), \cdots, (a_{\langle n \rangle}^0, y_{\langle n \rangle}) \right) \\
&= \Pr\left( [\![a_{\langle 1 \rangle}^1]\!], \cdots, [\![a_{\langle n \rangle}^1]\!] | (a_{\langle 1 \rangle}^1, y_{\langle 1 \rangle}), \cdots, (a_{\langle n \rangle}^1, y_{\langle n \rangle}) \right).
\end{aligned}
\tag{18}
$$

We will prove Eqn. (18) by induction on $n$. We first have

For $n = 1$, we have $[\![a^0_{\langle 1 \rangle}]\!] = c^0_{max}/2$ and $[\![a^1_{\langle 1 \rangle}]\!] = c^1_{max}/2$ with $c^0_{max} = c^1_{max} = 2^{\lambda \log_2 n}$, according to the initialization in Algorithm 1. This follows that

$$\Pr\left([\![a^0_{\langle 1 \rangle}]\!]|(a^0_{\langle 1 \rangle}, y_{\langle 1 \rangle})\right) = \Pr\left([\![a^0_{\langle 1 \rangle}]\!]\right) = \Pr\left([\![a^1_{\langle 1 \rangle}]\!]\right) = \Pr\left([\![a^1_{\langle 1 \rangle}]\!]|(a^1_{\langle 1 \rangle}, y_{\langle 1 \rangle})\right) .$$

We assume that Eqn. (18) holds for $n = i$ $(i > 1)$, that is,

$$\Pr\left([\![a^0_{\langle 1 \rangle}]\!], \cdots, [\![a^0_{\langle i \rangle}]\!]|(a^0_{\langle 1 \rangle}, y_{\langle 1 \rangle}), \cdots, (a^0_{\langle i \rangle}, y_{\langle i \rangle})\right)$$
$$= \Pr\left([\![a^1_{\langle 1 \rangle}]\!], \cdots, [\![a^1_{\langle i \rangle}]\!]|(a^1_{\langle 1 \rangle}, y_{\langle 1 \rangle}), \cdots, (a^1_{\langle i \rangle}, y_{\langle i \rangle})\right) . \tag{19}$$

Let us consider the case $n = i + 1$, and we add the sample $(a^b_{\langle i+1 \rangle}, y_{\langle i+1 \rangle})$ in binary search tree $\mathcal{BT}^b$ (Algorithm 1). It is sufficient to consider two cases as follows:

- If we do not need to split a node for sample $(a^b_{\langle i+1 \rangle}, y_{\langle i+1 \rangle})$ in Algorithm 1, then we have $[\![a^1_{\langle i+1 \rangle}]\!] = [\![a^0_{\langle i+1 \rangle}]\!]$. This is because $a^b_{\langle 1 \rangle} < a^b_{\langle 2 \rangle} < \cdots < a^b_{\langle i+1 \rangle}$ for $b = 0$ and $b = 1$, along with the same labels $\{y_1, \cdots, y_{i+1}\}$. Hence, we obtain the same ciphertext for $a^0_{\langle i+1 \rangle}$ and $a^1_{\langle i+1 \rangle}$, i.e., $t^0.cipher_1 = t^1.cipher_1$. This follows that

$$\Pr\left([\![a^0_{\langle 1 \rangle}]\!], \cdots, [\![a^0_{\langle i \rangle}]\!], [\![a^0_{\langle i+1 \rangle}]\!]|(a^0_{\langle 1 \rangle}, y_{\langle 1 \rangle}), \cdots, (a^0_{\langle i \rangle}, y_{\langle i \rangle}), (a^0_{\langle i+1 \rangle}, y_{\langle i+1 \rangle})\right)$$
$$= \Pr\left([\![a^0_{\langle 1 \rangle}]\!], \cdots, [\![a^0_{\langle i \rangle}]\!]|(a^0_{\langle 1 \rangle}, y_{\langle 1 \rangle}), \cdots, (a^0_{\langle i \rangle}, y_{\langle i \rangle})\right) ,$$

and

$$\Pr\left([\![a^1_{\langle 1 \rangle}]\!], \cdots, [\![a^1_{\langle i \rangle}]\!], [\![a^1_{\langle i+1 \rangle}]\!]|(a^1_{\langle 1 \rangle}, y_{\langle 1 \rangle}), \cdots, (a^1_{\langle i \rangle}, y_{\langle i \rangle}), (a^1_{\langle i+1 \rangle}, y_{\langle i+1 \rangle})\right)$$
$$= \Pr\left([\![a^1_{\langle 1 \rangle}]\!], \cdots, [\![a^1_{\langle i \rangle}]\!]|(a^1_{\langle 1 \rangle}, y_{\langle 1 \rangle}), \cdots, (a^1_{\langle i \rangle}, y_{\langle i \rangle})\right) .$$

By induction assumption in Eqn. (19), we have

$$\Pr\left([\![a^0_{\langle 1 \rangle}]\!], \cdots, [\![a^0_{\langle i \rangle}]\!], [\![a^0_{\langle i+1 \rangle}]\!]|(a^0_{\langle 1 \rangle}, y_{\langle 1 \rangle}), \cdots, (a^0_{\langle i \rangle}, y_{\langle i \rangle}), (a^0_{\langle i+1 \rangle}, y_{\langle i+1 \rangle})\right)$$
$$= \Pr\left([\![a^1_{\langle 1 \rangle}]\!], \cdots, [\![a^1_{\langle i \rangle}]\!], [\![a^1_{\langle i+1 \rangle}]\!]|(a^1_{\langle 1 \rangle}, y_{\langle 1 \rangle}), \cdots, (a^1_{\langle i \rangle}, y_{\langle i \rangle}), (a^1_{\langle i+1 \rangle}, y_{\langle i+1 \rangle})\right) .$$

- If we need to split the node for $(a^b_{\langle i+1 \rangle}, y_{\langle i+1 \rangle})$ in Algorithm 1, then we assume that $t^0$ and $t^1$ are the corresponding splitting nodes. We firstly initialize the empty node $l^b$ and $r^b$, and update the ciphertext $l^b.cipher_1$ and $r^b.cipher_1$ by Eqns (8)-(11), respectively. Notice that the random number $\xi$ in Eqns (8)-(11) is sampled from $\mathcal{N}(0, 1)$, and thus $l^0.cipher_1$ and $l^1.cipher_1$ are sampled from the same distribution. We have

$$\Pr\left(l^0.cipher_1|(a^0_{\langle 1 \rangle}, y_{\langle 1 \rangle}), \cdots, (a^0_{\langle i \rangle}, y_{\langle i \rangle}), (a^0_{\langle i+1 \rangle}, y_{\langle i+1 \rangle})\right)$$
$$= \Pr\left(l^1.cipher_1|(a^1_{\langle 1 \rangle}, y_{\langle 1 \rangle}), \cdots, (a^1_{\langle i \rangle}, y_{\langle i \rangle}), (a^1_{\langle i+1 \rangle}, y_{\langle i+1 \rangle})\right) .$$

Similarly, $r^0.cipher_1$ and $r^1.cipher_1$ are sampled from the same distribution, and we have

$$\Pr\left(r^0.cipher_1|(a^0_{\langle 1 \rangle}, y_{\langle 1 \rangle}), \cdots, (a^0_{\langle i \rangle}, y_{\langle i \rangle}), (a^0_{\langle i+1 \rangle}, y_{\langle i+1 \rangle})\right)$$
$$= \Pr\left(r^1.cipher_1|(a^1_{\langle 1 \rangle}, y_{\langle 1 \rangle}), \cdots, (a^1_{\langle i \rangle}, y_{\langle i \rangle}), (a^1_{\langle i+1 \rangle}, y_{\langle i+1 \rangle})\right) .$$

For the $(i + 1)$-th iteration in Algorithm 1, we have

$$\Pr\left([\![a^0_{\langle 1 \rangle}]\!], \cdots, [\![a^0_{\langle i \rangle}]\!], [\![a^0_{\langle i+1 \rangle}]\!]|(a^0_{\langle 1 \rangle}, y_{\langle 1 \rangle}), \cdots, (a^0_{\langle i \rangle}, y_{\langle i \rangle}), (a^0_{\langle i+1 \rangle}, y_{\langle i+1 \rangle})\right)$$
$$= \Pr\left([\![a^0_{\langle 1 \rangle}]\!], \cdots, [\![a^0_{\langle i \rangle}]\!]|(a^0_{\langle 1 \rangle}, y_{\langle 1 \rangle}), \cdots, (a^0_{\langle i \rangle}, y_{\langle i \rangle})\right)$$
$$\times \Pr\left(l^0.cipher_1|(a^0_{\langle 1 \rangle}, y_{\langle 1 \rangle}), \cdots, (a^0_{\langle i \rangle}, y_{\langle i \rangle}), (a^0_{\langle i+1 \rangle}, y_{\langle i+1 \rangle})\right)$$
$$\times \Pr\left(r^0.cipher_1|(a^0_{\langle 1 \rangle}, y_{\langle 1 \rangle}), \cdots, (a^0_{\langle i \rangle}, y_{\langle i \rangle}), (a^0_{\langle i+1 \rangle}, y_{\langle i+1 \rangle})\right) ,$$

**Table 4:** Hyperparameter settings for tree ensemble models in experiments. '–' means that the parameter is not exist in the corresponding method, and 'max_bin' denotes the maximum splitting point of each feature.

| Parameter | Our Work | PPD–ERTs | HEldpRFs | PivotRFs | MulPRFs | AnonyRFs | DiffPrivRFs | Original RFs |
|---|---|---|---|---|---|---|---|---|
| max_depth | None | None | 5 | 4 | None | None | None | None |
| n_estimators | 100 | 100 | 100 | 100 | 100 | 100 | 100 | 100 |
| max_features | $\lfloor\sqrt{d}\rfloor$ | $\lfloor\sqrt{d}\rfloor$ | $\lfloor\sqrt{d}\rfloor$ | $\lfloor\sqrt{d}\rfloor$ | $\lfloor\sqrt{d}\rfloor$ | $\lfloor\sqrt{d}\rfloor$ | $\lfloor\sqrt{d}\rfloor$ | $\lfloor\sqrt{d}\rfloor$ |
| differentia privacy level $\epsilon$ | – | – | – | – | – | – | 1 | – |
| anonymization parameter $k$ | – | – | – | – | – | 10 | – | – |
| multi–party size $p$ | 2 | 2 | 2 | 2 | 2 | – | – | – |
| max_bin | – | – | – | 16 | – | – | – | – |

and

$$\Pr\left(\llbracket a^1_{\langle 1\rangle}\rrbracket, \cdots, \llbracket a^1_{\langle i\rangle}\rrbracket, \llbracket a^1_{\langle i+1\rangle}\rrbracket | (a^1_{\langle 1\rangle}, y_{\langle 1\rangle}), \cdots, (a^1_{\langle i\rangle}, y_{\langle i\rangle}), (a^1_{\langle i+1\rangle}, y_{\langle i+1\rangle})\right)$$

$$= \Pr\left(\llbracket a^1_{\langle 1\rangle}\rrbracket, \cdots, \llbracket a^1_{\langle i\rangle}\rrbracket | (a^1_{\langle 1\rangle}, y_{\langle 1\rangle}), \cdots, (a^1_{\langle i\rangle}, y_{\langle i\rangle})\right)$$

$$\times \Pr\left(l^1.cipher_1 | (a^1_{\langle 1\rangle}, y_{\langle 1\rangle}), \cdots, (a^1_{\langle i\rangle}, y_{\langle i\rangle}), (a^1_{\langle i+1\rangle}, y_{\langle i+1\rangle})\right)$$

$$\times \Pr\left(r^1.cipher_1 | (a^1_{\langle 1\rangle}, y_{\langle 1\rangle}), \cdots, (a^1_{\langle i\rangle}, y_{\langle i\rangle}), (a^1_{\langle i+1\rangle}, y_{\langle i+1\rangle})\right) .$$

This follows that

$$\Pr\left(\llbracket a^0_{\langle 1\rangle}\rrbracket, \cdots, \llbracket a^0_{\langle i\rangle}\rrbracket, \llbracket a^0_{\langle i+1\rangle}\rrbracket | (a^0_{\langle 1\rangle}, y_{\langle 1\rangle}), \cdots, (a^0_{\langle i\rangle}, y_{\langle i\rangle}), (a^0_{\langle i+1\rangle}, y_{\langle i+1\rangle})\right)$$

$$= \Pr\left(\llbracket a^1_{\langle 1\rangle}\rrbracket, \cdots, \llbracket a^1_{\langle i\rangle}\rrbracket, \llbracket a^1_{\langle i+1\rangle}\rrbracket | (a^1_{\langle 1\rangle}, y_{\langle 1\rangle}), \cdots, (a^1_{\langle i\rangle}, y_{\langle i\rangle}), (a^1_{\langle i+1\rangle}, y_{\langle i+1\rangle})\right) .$$

This completes the proof. □

# D   Experimental Details

**Experimental settings**

We now present some details of compared methods in this work.

- **Original RFs**[3]: The orignal plaintext random forests [1] implemented by sklearn;
- **PPD-ERTs**[4]: The extremely randomized trees algorithm for learning from distributed horizontal partition data [64];
- **PivotRFs**[5]: A private and efficient solution for tree-based models in a vertical federated learning setting [62], based on a hybrid of threshold partially homomorphic encryption and secure multiparty computation techniques;
- **MulPRFs**[6]: The original random forest [1] with the secure multiparty computation library MP-SPDZ [94], based on the sh2 protocol to support semi-honest two-party computation;
- **AnonyRFs**[7]: The random forests based on anonymization library Mondrian, is a top-down greedy data anonymization algorithm for relational dataset [59];
- **DiffPrivRFs**[8]: Random forests based on differential privacy library Diffprivlib [93].

Tables 4 and 5 summarizes some hyperparameters settings in our experiments. Except for parameters 'n_estimators' and '$\alpha$' in leaf splitting, other parameters are set according to their respective references. We set security parameter $\lambda > 6.4$ according to privacy-preserving requisites as in [89].

---

[3] The code is downloaded from *github.com/scikit-learn/scikit-learn*

[4] The code is downloaded from *github.com/AminAminifar/kPPDERT_cloud*

[5] The code is downloaded from *github.com/nusdbsystem/pivot*

[6] The code is downloaded from *github.com/csiro-mlai/decision-tree-mpc*

[7] The code is downloaded from *github.com/qiyuangong/Mondrian*

[8] The code is downloaded from *github.com/IBM/differential-privacy-library*

**Table 5:** Hyperparameter setting of samples' minimum number $\alpha$ for leaves splitting in experiments.

| Parameter | wdbc | cancer | breast | diabetes | german | adver | bibtex | phpB0 | pendigits | phish |
|-----------|------|--------|--------|----------|--------|-------|--------|-------|-----------|-------|
| $\alpha$ | 10 | 10 | 10 | 10 | 10 | 10 | 10 | 10 | 10 | 10 |

| Parameter | ailerons | house | a9a | amazon | bank | adult | mnist | miniboone | runwalk | covtype |
|-----------|----------|-------|-----|--------|------|-------|-------|-----------|---------|---------|
| $\alpha$ | 10 | 100 | 100 | 100 | 100 | 100 | 100 | 100 | 100 | 100 |

**Table 6:** The orders of magnitude improvement compared to other approaches in Figure 2. 'NA' means that no results were obtained after running out $10^6$ seconds (about 11.6 days).

| Dataset | Our encrypted RFs | Original RFs | AnonyRFs | DiffPrivRFs | PPD-ERTs | PivotRFs | MulPRFs | HEldpRFs |
|---------|-------------------|--------------|----------|-------------|----------|----------|---------|----------|
| wdbc | 1 | $\times 10^{-3}$ | $\times 10^{-3}$ | $\times 10^{-3}$ | $\times 2$ | $\times 10$ | $\times 25$ | $\times 400$ |
| cancer | 1 | $\times 10^{-3}$ | $\times 10^{-3}$ | $\times 10^{-3}$ | $\times 1.5$ | $\times 10$ | $\times 20$ | $\times 300$ |
| breast | 1 | $\times 10^{-3}$ | $\times 10^{-3}$ | $\times 10^{-3}$ | $\times 2$ | $\times 13$ | $\times 30$ | $\times 10^3$ |
| german | 1 | $\times 10^{-3}$ | $\times 10^{-3}$ | $\times 10^{-3}$ | $\times 2$ | $\times 18$ | $\times 40$ | $\times 3000$ |
| diabetes | 1 | $\times 10^{-3}$ | $\times 10^{-3}$ | $\times 10^{-3}$ | $\times 2$ | $\times 15$ | $\times 25$ | $\times 850$ |
| adver | 1 | $\times 10^{-3}$ | $\times 10^{-3}$ | $\times 10^{-3}$ | NA | $\times 475$ | NA | NA |
| bibtex | 1 | $\times 10^{-3}$ | $\times 10^{-3}$ | $\times 10^{-3}$ | NA | $\times 328$ | NA | NA |
| phpB0 | 1 | $\times 10^{-3}$ | $\times 10^{-3}$ | $\times 10^{-3}$ | NA | NA | NA | NA |
| pendigits | 1 | $\times 10^{-4}$ | $\times 10^{-4}$ | $\times 10^{-4}$ | $\times 2$ | $\times 25$ | NA | NA |
| phish | 1 | $\times 10^{-3}$ | $\times 10^{-3}$ | $\times 10^{-3}$ | $\times 1$ | $\times 139$ | $\times 848$ | NA |
| ailerons | 1 | $\times 10^{-4}$ | $\times 10^{-4}$ | $\times 10^{-4}$ | $\times 1$ | $\times 31$ | $\times 40$ | NA |
| house | 1 | $\times 10^{-4}$ | $\times 10^{-4}$ | $\times 10^{-4}$ | $\times 1$ | $\times 31$ | $\times 38$ | NA |
| a9a | 1 | $\times 10^{-3}$ | $\times 10^{-3}$ | $\times 10^{-3}$ | $\times 1$ | $\times 453$ | $\times 762$ | NA |
| amazon | 1 | $\times 10^{-4}$ | $\times 10^{-4}$ | $\times 10^{-4}$ | $\times 1$ | $\times 51$ | $\times 31$ | NA |
| bank | 1 | $\times 10^{-4}$ | $\times 10^{-4}$ | $\times 10^{-4}$ | $\times 1.5$ | $\times 149$ | $\times 220$ | NA |
| adult | 1 | $\times 10^{-3}$ | $\times 10^{-3}$ | $\times 10^{-3}$ | $\times 2$ | $\times 211$ | $\times 276$ | NA |
| mnist | 1 | $\times 10^{-4}$ | $\times 10^{-4}$ | $\times 10^{-4}$ | NA | NA | NA | NA |
| miniboone | 1 | $\times 10^{-4}$ | $\times 10^{-4}$ | $\times 10^{-4}$ | $\times 2$ | $\times 35$ | NA | NA |
| runwalk | 1 | $\times 10^{-4}$ | $\times 10^{-4}$ | $\times 10^{-4}$ | $\times 2$ | $\times 35$ | NA | NA |
| covtype | 1 | $\times 10^{-3}$ | $\times 10^{-3}$ | $\times 10^{-3}$ | $\times 1$ | NA | NA | NA |

## Running Time

We give the prediction time comparisons(in seconds) for different methods as shown in Figure 4. As we can see, our encrypted random forests take comparable running time with original random forests, AnonyRFs and DiffPrivRFs, since our Gini-impurity preserving encryption method only requires $O(h)$ time complexity without other additional operations, where $O(h)$ denotes the height of binary search tree $\mathcal{BT}$.

Furthermore, our encrypted random forests show superior efficiency compared to other methods, such as MulPRFs, PPD-ERTs, PivotRFs, and HEldpRFs, with the training time obtained in $10^6$ seconds (almost 11.6 days). This is because MulPRFs, PivotRFs, and PPD-ERTs require expensive communication costs for multi-parity computation, while HEldpRFs takes heavy computation costs on HE scheme. We also present the orders of magnitude improvement of training and prediction time in Table 6 and Table 7, respectively.

## Security

We analyze the security across fourteen datasets by randomly selecting an attribute that share a similar trend as other dimensions. We compare our Gini-impurity-preserving scheme with other four privacy-protection methods: differential privacy [93], anonymization [59], order-preserving scheme [96] and HE scheme [42]. The results are depicted in Figure 5.

Inspired from [97], we take the bitwise leakage matrix as our metric. An initial step is to scale and discretize the feature space into integers within the range of $[0, 2^7]$, and then we sample 200 representative samples from each dataset to evaluate the security of the feature space. The primary objective in experiment is to safeguard as many bits of the plaintexts as possible. This quantitative

**Table 7:** The orders of magnitude improvement compared to other approaches in Figure 4. 'NA' means that no results were obtained after running out $10^6$ seconds (about 11.6 days).

| Dataset | Our encrypted RFs | Original RFs | AnonyRFs | DiffPrivRFs | PPD-ERTs | PivotRFs | MulPRFs | HEldpRFs |
|---|---|---|---|---|---|---|---|---|
| wdbc | 1 | ×3 | ×3 | ×3 | ×38 | ×1,220 | ×93 | ×4,000 |
| cancer | 1 | ×28 | ×29 | ×25 | ×360 | ×11,052 | ×851 | ×41,911 |
| breast | 1 | ×25 | ×31 | ×27 | ×308 | ×11,631 | ×776 | ×44,736 |
| german | 1 | ×4 | ×5 | ×4 | ×115 | ×2,615 | ×421 | ×9,615 |
| diabetes | 1 | ×42 | ×35 | ×41 | ×411 | ×18,142 | ×1,642 | ×64,285 |
| adver | 1 | ×3 | ×4 | ×10 | NA | ×3,821 | NA | NA |
| bibtex | 1 | ×1 | ×1 | ×1 | NA | ×1,528 | NA | NA |
| phpB0 | 1 | ×4 | ×4 | ×4 | NA | NA | NA | NA |
| pendigits | 1 | ×6 | ×6 | ×10 | ×2384 | ×18,947 | NA | NA |
| phish | 1 | ×5 | ×8 | ×6 | ×1,966 | ×1,7619 | ×2,604 | NA |
| ailerons | 1 | ×6 | ×9 | ×8 | ×1,581 | ×30,200 | ×3,600 | NA |
| house | 1 | ×6 | ×9 | ×8 | ×1,581 | ×30,400 | ×3,600 | NA |
| a9a | 1 | ×6 | ×10 | ×8 | ×5,482 | ×27,000 | ×3,250 | NA |
| amazon | 1 | ×10 | ×12 | ×12 | ×2,208 | ×54,500 | ×7,500 | NA |
| bank | 1 | ×14 | ×18 | ×20 | ×5,637 | ×75,500 | ×10,000 | NA |
| adult | 1 | ×4 | ×5 | ×4 | ×1,967 | ×22,054 | ×2,876 | NA |
| mnist | 1 | ×2 | ×3 | ×2 | NA | NA | NA | NA |
| miniboone | 1 | ×6 | ×9 | ×9 | ×1,800 | ×75,000 | NA | NA |
| runwalk | 1 | ×12 | ×18 | ×26 | ×2,413 | ×84,000 | NA | NA |
| covtype | 1 | ×7 | ×10 | ×8 | ×2,943 | NA | NA | NA |

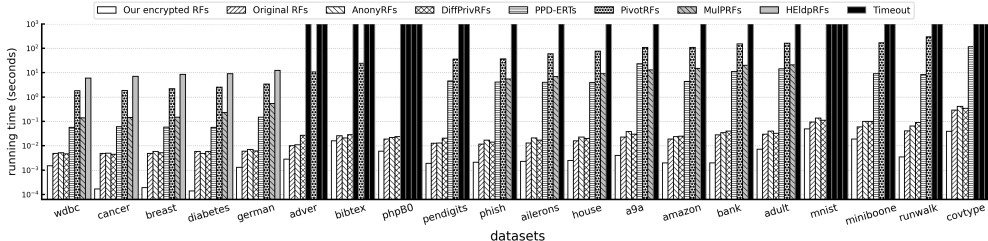

**Figure 4:** Comparisons of the prediction running time on different random forest. Notice that the y-axis is in log-scale, and full black columns imply that no result was obtained after running out $10^6$ seconds for training (about 11.6 days).

assessment is visualized through a color map: the $x$-axis represents the individual bits (1 through 7), while the $y$-axis indicates the rank order of the 200 sampled datasets.

The color gradient, ranging from white to red, represents the degree of security, with white correlating to minimal security and red to maximal security. The security degree was normalized within a $[0, 1]$ range to ensure results' consistency. For instance, a security degree of 0 with white color indicates no security, while a security degree of 1 with red color suggests the highest level of security.

As expected, the HE scheme presents the highest security, yet with heavy computational costs. For example, datasets exceeding 3000 samples yielded no results under the HE scheme, even with an extended runtime of $10^6$ seconds. It is also observed that our proposed scheme demonstrated superior security efficacy compared to the other three scheme: differential privacy [93], anonymization [59] and order-preserving scheme [96]. Since those schemes rely on mere data perturbations, compressions, or order information preservation. In comparison, our scheme makes a good balance between security and computational cost.

# E    Proof of Bitwise Leakage

In this section, we present a comprehensive evaluation of the security properties for our Gini-impurity preserving methods, full homomorphic encryption, anonymization technique, and differential privacy

methods. The security analysis is conducted in the feature space using the bitwise leakage matrix which is proposed by [97].

We focus on a discrete and finite feature space with a fixed size as in [97]. The feature space is defined as $\mathcal{X} = [0, 2^{m-1}]$, which means that the feature size is $m$ bits, and the space ranges from 0 to $2^{m-1}$. Let $\mathcal{D}$ be the true distribution over the feature space, and dataset $S = \{a_1, \ldots, a_n\}$ are sampled independently and identically from distribution $\mathcal{D}$.

The adversary $\mathcal{A}$ possesses two types of knowledge to achieve the goal of recovering plaintexts:

- Auxiliary knowledge about a distribution $\mathcal{D}'$ over the feature space $\mathcal{X}$ [98], which provides additional information to the adversary.
- Ciphertexts $[\![S]\!]$ corresponding to $S$, which represents the snapshot of the encrypted data store, as described in Fuller et al. [99].

We re-sort dataset $S$ with a non-decreasing order, i.e., $S = \{a_{\langle 1 \rangle}, a_{\langle 2 \rangle}, \cdots, a_{\langle n \rangle}\}$ where $a_{\langle 1 \rangle} \leq a_{\langle 2 \rangle} \leq \cdots \leq a_{\langle n \rangle}$. Let $S\langle i \rangle$ be the $i$-th sample in $S$, and $S\langle i \rangle_{[j]}$ be the $j$-th bit of $S\langle i \rangle$ with $i \in [n]$ and $j \in [m]$. Then, we denote by $b\langle i \rangle_{[j]}$ the adversary's guess for $S\langle i \rangle_{[j]}$ through the auxiliary knowledge distribution $\mathcal{D}'$ as follows:

$$b\langle i \rangle_{[j]} = \arg \max_{b \in \{0,1\}} \Pr_{\mathcal{D}'}\left(S\langle i \rangle_{[j]} = b\right) = \left\{ \begin{array}{lll} 0 & \text{for} & \mathbb{E}_{\mathcal{D}'}[S\langle i \rangle_{[j]}] \leq 1/2 \\ 1 & \text{for} & \mathbb{E}_{\mathcal{D}'}[S\langle i \rangle_{[j]}] > 1/2 \,, \end{array} \right.$$

for $i \in [n]$ and $j \in [m]$. The adversary aims to correctly guess the plaintext $S\langle i \rangle_{[j]}$ using the auxiliary knowledge $\mathcal{D}'$. Let $\mathcal{L}$ be a $n \times m$ matrix with

$$\mathcal{L}(i, j) = \Pr\left(S\langle i \rangle_{[j]} = b\langle i \rangle_{[j]} | \mathcal{D}, \mathcal{D}'\right) \text{ for } i \in [n] \text{ and } j \in [m] \,.$$

Similarly to [97], we have

$$\Pr_{\mathcal{D}}\left(S\langle i \rangle_{[j]} = 0\right) = \sum_{s \in S_0^j} \Pr_{\mathcal{D}}\left(S\langle i \rangle = s\right) \text{ for } i \in [n] \text{ and } j \in [m] \,,$$

and $\Pr_{\mathcal{D}}\left(S\langle i \rangle_{[j]} = 1\right) = 1 - \Pr_{\mathcal{D}}\left(S\langle i \rangle_{[j]} = 0\right)$, where $s_{[j]}$ denotes the $j$-th bit of $s$ with $S_0^j = \{s | s \in \mathcal{S} \text{ and } s_{[j]} = 0\}$. This follows that

$$\mathcal{L}(i, j) = \Pr\left(S\langle i \rangle_{[j]} = b\langle i \rangle_{[j]} | \mathcal{D}, \mathcal{D}'\right) = \sum_{s \in S_{b\langle i \rangle_{[j]}}^j} \Pr_{\mathcal{D}}\left(S\langle i \rangle = s\right) \,.$$

The variable $\mathcal{L}(i, j)$ represents the probability that an adversary can accurately guess the $j$-th bit of the plaintext $S\langle i \rangle$. This metric can be considered as a measure of the information security for the ciphertexts $[\![S\langle i \rangle]\!]$, in the sense that a lower value of $\mathcal{L}(i, j)$ signifies a higher degree of security. Specifically, the bitwise information security of $[\![S\langle i \rangle]\!]$ can be quantified as 1-$\mathcal{L}(i, j)$, and this metric provides a precise and quantitative assessment of the encryption scheme's security properties.

Specifically, we investigate the correlation among elements of $\mathcal{L}(i, j)$, plaintexts, ciphertexts and secret keys. We explore the impact of different encryption parameters on the structure and behavior of $\mathcal{L}(i, j)$. Our analysis reveals that the leakage pattern of $\mathcal{L}(i, j)$ is highly dependent on the specific encryption scheme. Therefore, it is crucial to carefully design and select the appropriate encryption scheme to minimize the risk of information leakage.

We now present the analysis of bitwise leakage matrix $\mathcal{L}(i, j)$ for our encryption method as follows.

**Theorem 7.** *For our Gini-impurity-preserving encryption and plaintexts $S$, we have*

$$L(i, j) = \sum_{q \in [i, n-k+i]} \frac{\mathbb{I}(S\langle q \rangle_{[j]}) = S\langle i \rangle_{[j]})}{n - k + 1} \times \sum_{s \in S_{b\langle i \rangle_{[j]}}^j} \Pr_D\left(S\langle i \rangle = s\right) + \textit{small constant} \,.$$

*Proof.* Our Gini-impurity-preserving encryption transfers multiple plaintexts in $\mathcal{I}_{i'}$ ($i' \in [k]$) to the identical first dimension ciphertext, i.e., $c_{i'}$, as shown in Eqn. (5). Hence, the $i'$-th ciphertext $c_{i'}$ corresponds to multiple plaintexts, and the adversary will guess the true plaintext $S\langle i \rangle$ of ciphertext

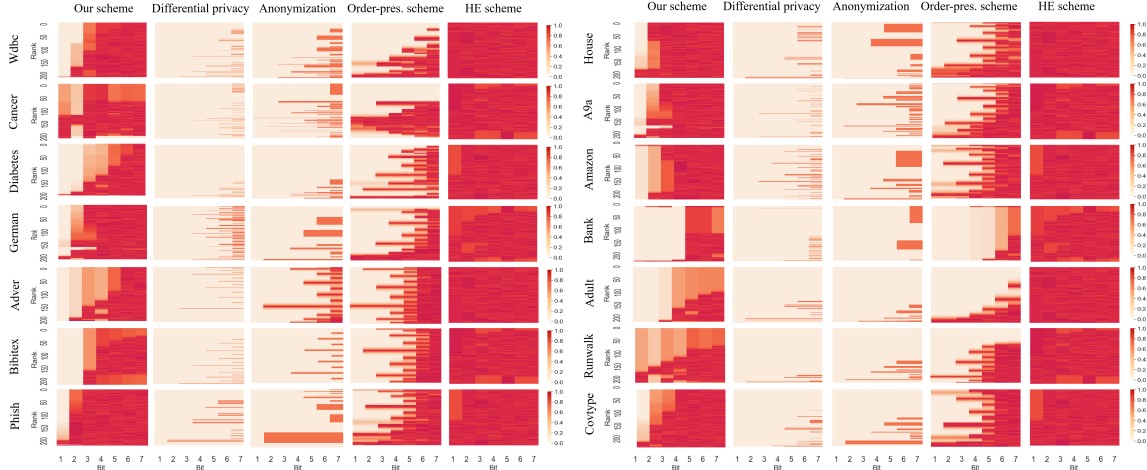

**Figure 5:** Comparisons of the security degree for the feature space through the bitwise leakage matrix.

$c_{i'}$. Since the adversary only knows that $i' - 1$ ciphertexts are smaller than $c_{i'}$ and $k - i'$ ciphertexts are larger than $c_{i'}$, the adversary can guess the plaintext $S\langle i\rangle$ from

$$\{S\langle q\rangle | q \in [i, n-k+i]\}$$

with the same probability. In this way, the probability of adversary guessing $S\langle i\rangle_{[j]}$ is

$$P_{i,j} = \sum_{q \in [i, n-k+i]} \frac{\mathbb{I}(S\langle q\rangle_{[j]} = S\langle i\rangle_{[j]})}{n-k+1} \ .$$

Let $b\langle i\rangle_{[j]}$ be the adversary's guess for $S\langle i\rangle_{[j]}$, we have

$$b\langle i\rangle_{[j]} = \arg\max_{b \in \{0,1\}} \Pr_{\mathcal{D}'}\left(S\langle i\rangle_{[j]} = b\right) = \begin{cases} 0 & \text{for} & \mathbb{E}_{\mathcal{D}'}[S\langle i\rangle_{[j]}] \le 1/2 \\ 1 & \text{for} & \mathbb{E}_{\mathcal{D}'}[S\langle i\rangle_{[j]}] > 1/2 \ . \end{cases}$$

The probability for the adversary correctly identifies the $j$-th bit of the plaintext $S\langle i\rangle$ is

$$L(i,j) = P_{i,j} \sum_{s \in S_{b\langle i\rangle_{[j]}}^{j}} \Pr_{\mathcal{D}}\left(S\langle i\rangle = s\right) + \text{ small constant },$$

and we complete the proof from Lemma 8. $\qquad\square$

**Lemma 8** (Roy et al. [97])**.** *Let $\mathcal{D}$ be the input distribution and $S = \{a_1, \ldots, a_n\}$ denotes the dataset with each data point sampled i.i.d. from $\mathcal{D}$, then we have*

$$\Pr_{\mathcal{D}}\left(S\langle i\rangle = a'\right) = \sum_{j=n-i+1}^{n} \binom{n}{j} \left(\Pr_{\mathcal{D}}\left(a < a'\right)\right)^{n-j} \left(\Pr_{\mathcal{D}}\left(a = a'\right)\right)^{j} \ \text{ for } \ \Pr_{\mathcal{D}}\left(a > a'\right) = 0 \ ,$$

*and*

$$\Pr_{\mathcal{D}}\left(S\langle i\rangle = a'\right) = \sum_{j=i}^{n} \binom{n}{j} \left(\Pr_{\mathcal{D}}\left(a = a'\right)\right)^{j} \left(\Pr_{\mathcal{D}}\left(a > a'\right)\right)^{n-j} \ \text{ for } \ \Pr_{\mathcal{D}}\left(a < a'\right) = 0 \ ;$$

*otherwise,*

$$\Pr_{\mathcal{D}}\left(S\langle i\rangle = a'\right) = \sum_{j=1}^{n} \sum_{k=\max\{1, i-j+1\}}^{\min\{i, n-j+1\}} \binom{n}{k-1, j, n-k-j+1} \Delta_{k-1, j, n-k-j+1} \ ,$$

*where*

$$\Delta_{k-1, j, n-k-j+1} = \left(\Pr_{\mathcal{D}}\left(a < a'\right)\right)^{k-1} \cdot \left(\Pr_{\mathcal{D}}\left(a = a'\right)\right)^{j} \cdot \left(\Pr_{\mathcal{D}}\left(a > a'\right)\right)^{n-k-j+1} \ .$$

We now provide similar analysis of bitwise leakage matrix $\mathcal{L}$ for $\epsilon$-local differential privacy.

**Theorem 9.** *For $\epsilon$-local differential privacy, we have*

$$L(i,j) = \frac{\Pr\left(S\langle i\rangle_{[j]} = b\langle i\rangle_{[j]}\right) + \Pr\left(S\langle i\rangle_{[j]} = S'\langle i\rangle_{[j]}\right)}{2} + \text{ small constant },$$

*where $S$ and $S'$ denotes the plaintexts and ciphertexts, respectively, and*

$$b\langle i\rangle_{[j]} = \arg\max_{b\in\{0,1\}} \Pr_{\mathcal{D}'}\left(S\langle i\rangle_{[j]} = b\right) = \begin{cases} 0 & \text{if } \mathbb{E}_{\mathcal{D}'}[S\langle i\rangle_{[j]}] \leq 1/2 , \\ 1 & \text{if } \mathbb{E}_{\mathcal{D}'}[S\langle i\rangle_{[j]}] > 1/2 . \end{cases}$$

*Proof.* We concern $\epsilon$-local differential privacy by adding noise to each individual value. If the adversary attempts to infer the original plaintext $S\langle i\rangle$, then it relies on the $\epsilon$-differential privacy disturbed data $S'(i)$ and the auxiliary knowledge distribution $\mathcal{D}'$. The adversary guesses the $j$-th bit of the $i$-th plaintext $S\langle i\rangle_{[j]}$ through a process of deduction as follows:

$$S\langle i\rangle_{[j]} = \begin{cases} 1 & \text{for } b\langle i\rangle_{[j]} = 1 \text{ and } S'\langle i\rangle_{[j]} = 1 , \\ 0 & \text{for } b\langle i\rangle_{[j]} = 0 \text{ and } S'\langle i\rangle_{[j]} = 0 , \\ \text{randomly select from}\{0,1\} & \text{otherwise} . \end{cases}$$

This follows that

$$L(i,j) = \frac{\Pr\left(S\langle i\rangle_{[j]} = b\langle i\rangle_{[j]}\right) + \Pr\left(S\langle i\rangle_{[j]} = S'(i,j)\right)}{2} + \text{ small constant} .$$

This completes the proof. $\qquad\square$

In order to gain a deeper understanding of the security for the $k$-anonymous algorithm, we conduct an analysis of the bitwise leakage matrix $\mathcal{L}$. This matrix represents the amount of information leakage that occurs when the original data $X$ is compressed into $m$ partitions $\mathcal{K}_1, \mathcal{K}_2, \cdots, \mathcal{K}_t$ by the $k$-anonymous algorithm as follows:

$$\begin{aligned} \mathcal{K}_1 &= \left\{a_{\langle 1\rangle}, a_{\langle 2\rangle}, \cdots, a_{\langle k_1\rangle}\right\} \\ \mathcal{K}_2 &= \left\{a_{\langle k_1+1\rangle}, a_{\langle k_1+2\rangle}, \cdots, a_{\langle k_2\rangle}\right\} \\ &\cdots \\ \mathcal{K}_t &= \left\{a_{\langle k_{t-1}+1\rangle}, a_{\langle k_{t-1}+2\rangle}, \cdots, a_{\langle n\rangle}\right\} . \end{aligned}$$

The bitwise leakage matrix $\mathcal{L}$ quantifies the amount of information that can be inferred about an individual from the corresponding partitions. By analyzing this matrix, we can determine the level of privacy that is maintained by the $k$-anonymous algorithm and identify any potential vulnerabilities that could be exploited by an adversary. Then, we give the analysis of bitwise leakage matrix $\mathcal{L}$ for $k$-anonymous algorithm as follows.

**Theorem 10.** *For $k$-anonymous algorithm and the plaintexts $S$, we have*

$$L(i,j) = \Pr\left(S\langle i\rangle_{[j]} = b\langle i\rangle_{[j]}\right) + \text{small constan} ,$$

*where $S\langle i\rangle_{[j]}$ denotes the $j$-th bit of $S\langle i\rangle$ with $S\langle i\rangle \in \mathcal{K}_q(q \in [t])$ and*

$$b\langle i\rangle_{[j]} = \arg\max_{b\in\{0,1\}} \left\{\sum_{x\in\mathcal{K}_q} \mathbb{I}[x_{[j]} = b]\Pr_{\mathcal{D}'}(x)\right\} .$$

*Proof.* The $k$-anonymity is a privacy-preserving technique that aims to protect the identity of individuals in a dataset. It works by grouping together individuals with similar attributes and pooling their data in a larger group, thus making it difficult for an adversary to identify any specific individual in the group. The $k$-anonymity ensures that each group has at least $k$ individuals with the same attribute values, which further enhances the security of the data.

When the original data $S\langle i\rangle$ is pooled in the group $\mathcal{K}_q(q \in [t])$, the adversary attempts to guess the $j$-th bit of the plaintext $S\langle i\rangle$ using the auxiliary knowledge distribution $\mathcal{D}'$ and $\mathcal{K}_q$. To achieve this,

the adversary guesses $b\langle i\rangle_{[j]}$ as the value corresponding to the maximum probability of the $j$-th bit in group $\mathcal{K}_q$ as follows:

$$b\langle i\rangle_{[j]} = \arg \max_{b \in \{0,1\}} \left\{ \sum_{x \in \mathcal{K}_q} \mathbb{I}[x_{[j]} = b] \Pr_{\mathcal{D}'}(x) \right\} .$$

This completes the proof. $\qquad\square$

