# OpenReview forum: "On the Gini-impurity Preservation For Privacy Random Forests"
_NeurIPS.cc/2023/Conference — NeurIPS 2023 spotlight_

### Official Review · Reviewer_PTBU · 2023-07-02

**Soundness:** 1 poor
**Presentation:** 2 fair
**Contribution:** 3 good
**Rating:** 6
**Confidence:** 3

**Summary:**

This paper presents a novel encryption mechanism for isolation forest algorithms that preserves the Gini impurity metric. The authors provide theoretical evidence demonstrating that the proposed mechanism effectively preserves the Gini impurity and withstands attacks. Additionally, the authors have conducted comprehensive experiments to evaluate the accuracy and security of the proposed mechanism.

**Strengths:**

This paper presents novel and intriguing findings. The proposed encryption mechanism is innovative, and the experiments conducted seem to be extensive and thorough. Additionally, the experimental results demonstrate the promising performance of the proposed mechanism in terms of both accuracy and security.

**Weaknesses:**

1.  It appears that there are concerns regarding the solidity of the proof for the theoretical guarantee in the security analysis (Theorem 3). After reviewing the proof provided in the appendix, it is evident that the proof relies on experimental analysis utilizing different random seeds rather than a traditional mathematical proof. This raises questions about the rigorousness of the proof and its validity.

2. The paper requires improvement in terms of writing quality. In addition to the listed typos and grammar mistakes below, it is advisable for the authors to provide more background knowledge to enhance the reader's understanding. Furthermore, I suggest that the authors explain the motivation behind the proposed mechanism (Equations (4) and (5)) using a more informal language to aid comprehension.

I have noticed some typos and grammar mistakes in the abstract and introduction sections. However, I also observed grammar errors throughout the entire paper.
  1. Line 1, "algorithms" should be ``algorithm".

  2. Line 3, "from anonymization" might be "such as anonymization".

  3. Line 4, "it rarely takes into account" should be changed to "they rarely take into account".

  4. Line 8, "encrypt data features" should be changed to "encrypt the data features".

  5. Line 14, "effectiveness, efficiency and security" should be changed to "effectiveness, efficiency, and security".

  6. In the first sentence (Line 16), "one successful ensemble algorithms" should be ``one successful ensemble algorithm".

  7. Line 30, "cryptosystems" should be "cryptosystem".

  8. Line 22, "ingredients" should be "ingredient".

  9.  Line 61, "Given encryption function" should be "Given an encryption function". "and decryption function" should be "and a decryption function". "HE scheme" should be "the HE scheme"

**Questions:**

Based on the concerns regarding the unsolid theoretical guarantee and unclear writing, it seems that the current version of the paper may not be suitable for publication in NeurIPS. While the paper presents interesting results, addressing these issues is crucial for ensuring the quality and suitability of the paper for a prestigious conference like NeurIPS. It is recommended that the authors revise and improve the theoretical guarantee and clarity of the writing before considering submission to NeurIPS or any other reputable publication venue.

**Limitations:**

Yes.

---

> ### Author Rebuttal · Authors · 2023-08-06
>
> [Q1] … regarding the solidity of the proof for the theoretical guarantee in the security analysis (Theorem 3) … the proof relies on experimental analysis utilizing different random seeds rather than a traditional mathematical proof. This raises questions about the rigorousness of the proof and its validity.
>
> [A1] We will clarify that Theorem 3’ proof follows the cryptographic framework (Kerschbaum et al., 2015), which shows that Algorithm 1 yields the same ciphertext for any random seed and for any plaintext sequences {a_i^0} and {a_i^1}, which does not rely on experimental analysis. We can also present a traditional mathematical proof with rigorousness, inspired by Popa et al. (2013). The basic idea is to prove that, by induction on n,
> $P({[a_0^0] ,…,[a_i^0]}|{a_0^0,...,a_i^0}) =P({[a_0^1],…,[a_i^1]}|{a_0^1,...,a_i^1}) &ensp;for &ensp;i=0…n$,
> and then restrict an attacker's ability of accurate guess to a chance of success no more than 1/2. Here, $[a]$ denotes the ciphertext of $a$.
>
>
> [Q2]  I suggest that the authors explain the motivation behind the proposed mechanism (Equations (4) and (5)) …
>
> [A2] We will clarify that our motivation is that the Gini impurity does not changed by encrypting adjacent plaintexts with the same label into an identical value, which can be shown in Theorem 1. Hence, Eqn. (4) sorts and partitions plaintexts into different groups based on label information, and Eqn. (5) encodes different values within each group into the same ciphertext.
>
> We will improve the writing quality, and provide more background knowledge according to your suggestions. Thank you.

---

> > ### Comment · Reviewer_PTBU · 2023-08-13
> >
> > Thank you for your detailed comments. I would like to improve my rating to a 5, as I still have some concerns regarding your proof of Theorem 3.

---

> > > ### Author Response · Authors · 2023-08-15
> > >
> > > Dear Reviewer PTBU,
> > >
> > > We have presented  a traditional mathematical proof for Theorem 3, which is partially motivated from (Popa et al. 2013). We will add the detailed proof in Appendix, and give some proof sketches as follows:
> > >
> > > &nbsp;
> > >
> > > Let ${a_1^0,…,a_n^0}$ and ${a_1^1,…,a_n^1}$ be two sorted sequences (plaintexts) in an ascending order,  with corresponding labels ${y_1…,y_n}$ drwan i.i.d. from a uniform distribution.
> > >
> > > Theorem 3 follows if we can prove that, by induction on $n$,
> > >
> > > $P([a_1^0],…,[a_i^0] | (a_1^0,y_1),…,(a_i^0,y_i ))= P([a_1^1],…,[a_i^1] | (a_1^1,y_1),…,(a_i^1,y_i ))  \quad \quad \text{ for }i=1,...,n \quad \quad             (\text{I})$
> > >
> > > where $[a]$ denotes the corresponding ciphertext of plaintext $a$.
> > >
> > > &nbsp;
> > >
> > > For $n=1$, It is easy to prove $P([a_1^0]=c_{\max}/2| (a_1^0,y_1))= P([a_1^1]=c_{\max}/2 | (a_1^1,y_1))=1$ and Eqn. (I) holds obviously, since $[a_1^0]= [a_1^1]=c_{\max}/2$ from Algorithm 1, where $c_\max$ is a given large number as done in (Kerschbaum et al., 2015).
> > >
> > > &nbsp;
> > >
> > > We assume that Eqn. (I) holds for $n=i$ $(i>1)$, and we will prove the case $n=i+1$. It suffices to consider two cases:
> > >
> > > i)  If we do not require to split a node for data points $(a_{i+1}^b, y_{i+1}) (b=0,1)$, then binary search trees (constructed) have the same structure for $b=0$ and $b=1$ by induction assumption. We have $[a_{i+1}^0]=[a_{i+1}^1]$ from the same binary search trees and consistent order for plaintexts, which proves Eqn. (I) for $n=i+1$.
> > >
> > > ii) If we require to split a node for data points $(a_{i+1}^b, y_{i+1}) (b=0,1)$, then data points $(a_{i+1}^0,y_{i+1})$ and $(a_{i+1}^1,y_{i+1})$ fall into the corresponding node in binary search trees from Algorithm 1 (Section 3.2). According to Algorithm 2 (Section 3.2), we obtain new nodes  $l^0$, $r^0$ and $l^1$, $r^1$. By  induction assumption, the adversary obtains the same information for $b=0$ and $b=1$ when $n\leq i$, and hence ciphertexts of node $l^0$ and $l^1$ follow the same distribution.  We have
> > >
> > > $P( l^0.cipher_1 |  (a_1^0,y_1),…,(a_i^0,y_i),(a_{i+1}^0,y_{i+1} ) = P( l^1.cipher_1 |  (a_1^1,y_1),…,(a_i^1,y_i),(a_{i+1}^1,y_{i+1} ) $
> > >
> > > We could obtain similar result for $r^0$ and $r^1$. For $n=i+1$ and $b=0,1$, we finally have
> > >
> > > $P( [a_1^b],…,[a_i^b],[a_{i+1}^b] | (a_1^b,y_1),…,(a_i^b,y_i ),(a_{i+1}^b,y_{i+1})) = P( [a_1^b],…,[a_i^b] | (a_1^b,y_1),…,(a_i^b,y_i ) ) * P( l^b.cipher_1 |  (a_1^b,y_1),…,(a_i^b,y_i), (a_{i+1}^b,y_{i+1}) * P( r^b.cipher_1 |  (a_1^b,y_1),…,(a_i^b,y_i),(a_{i+1}^b,y_{i+1} ) , $
> > >
> > > which proves Eqn. (I) with some subsititutions.

---

### Official Review · Reviewer_B6q6 · 2023-07-07

**Soundness:** 3 good
**Presentation:** 3 good
**Contribution:** 2 fair
**Rating:** 6
**Confidence:** 1

**Summary:**

This work develops a new encryption to maintain the data's Gini impurity and combines it with CKKS to perform random forests training and predicting, which outperforms other SOTAs in terms of the trade-off between latency and security level.

**Strengths:**

1. This paper gets solid technique details and corresponding theoretical proof.
2. The author gives extensive experiment results and a wide range of datasets which shows the scalability of the proposed methods.

**Weaknesses:**

1. One weakness would be the limited application scenario. It would be better to have some variants for other applications.

**Questions:**

same with weakness

**Limitations:**

same with weakness

---

> ### Author Rebuttal · Authors · 2023-08-06
>
> [Q1] One weakness would be the limited application scenario. It would be better to have some variants for other applications.
>
> [A1] We will clarify that our work takes the first step on encryption by incorporating some learning ingredient, i.e., Gini impurity, and we can present some variants such as gini index and information gain, where the key idea is to show the piecewise monotonicity of information statistics w.r.t. splitting point. Those schemes can be applied to more learning scenarios such as Boosting, GBDT, LightGBT, etc.
>
> We will improve this work according to your suggestions. Thank you.

---

### Official Review · Reviewer_1ntL · 2023-07-09

**Soundness:** 3 good
**Presentation:** 3 good
**Contribution:** 3 good
**Rating:** 7
**Confidence:** 4

**Summary:**

This work investigates the privacy-preserving random forest, and the main contributions include an interesting property-preserving encryption method that can preserve data’s Gini impurity. Based on the proposed encryption method, the authors present an innovative privacy-preserving training and prediction algorithm for random forests in a client-server protocol, which can achieve the smallest communication and computational complexities in comparison with prior works. The method is supported by theoretical results, which provide justifications to preserve data’s Gini impurity and showed that the proposed algorithm is secure against Gini-impurity-preserving chosen plaintext attack. Finally, extensive experiments on various datasets are conducted to demonstrate the efficacy, efficiency, and security of the proposed method.

**Strengths:**

This paper is well-written and easy to follow. It provides novel algorithms and solid theories with encouraging empirical results. The main idea of designing new encryption algorithm for decision trees that can preserve minimum Gini impurity is interesting and convincing for me.

- Novel perspective on privacy-preserving machine learning. This paper introduces a novel perspective on privacy-preserving random forest by focusing on preserving minimum Gini impurity in encryption, since Gini impurity is exactly the information that random forests require during training and inference. This sheds new insights into the design of privacy-preserving machine learning algorithms with corresponding new property-preserving encryption methods, which is worthy of systemic further study for the potential enhancement of the efficacy and accuracy of privacy computing.
- Innovative solution with solid analysis. The idea of the proposed minimum Gini impurity preserving encryption is innovative and interesting. It uses a new binary search tree to encrypt data dynamically by incorporating labels’ information, which is a novel way to encrypt data for decision trees, and the proposed encryption method incurs little additional computational cost.  The communication complexity, security and the correctness of proposed method are also well-analysed.
- Well-founded experiments results. The empirical study provide accuracy and running time comparisons with various relevant methods, as well as a visual analysis of security. The encouraging result shows that the proposed method can effectively achieve a better balance among accuracy, communication, running time and data security.


**Weaknesses:**

Despite many strengths, there are several points where certain details could be further improved.

-	The paragraph about ``Gini-impurity-preserving chosen plaintext attack’’ feels somewhat abstract. As it relates to the security analysis of proposed method, additional explanation could help reduce the reading difficulty for readers. Including a more detailed explanation in appendix might make this critical point more accessible.
-	The interpretation of Figure 3 and the bitwise leakage matrices is a little confusing to me. It is not very clear to me what is the intuition of such matrices. It may be beneficial to provide a brief introduction on it to help readers comprehend Figure 3 more easily.
-	The introduction about homomorphic encryption and CKKS seems to be brief. More details may be provided in appendix for readers unfamiliar with homomorphic encryption.
-	In Figure 6 from the appendix, the ``Differential Privacy’’ at the top is blocked by the figure of bitwise leakage matrices.


**Questions:**

-	Homomorphic encryption is used to encrypt labels, and it is known that homomorphic encryption has a high computation overhead; does this impact the efficacy of the proposed method?
-	As shown in Table 3, the proposed encrypted random forests perform better than the original random forests on two datasets. Can you provide explanation for the improved efficacy on these datasets?
-	Can the proposed Gini-impurity-preserving encryption method be extended to Gini index or information gain?
-	In Algo. 1 , will the selection of $c_{max}$ impact the security of proposed method?


**Limitations:**

The authors have mentioned the limitations in security of the proposed method, such as the encryption method is secure against Gini-impurity-preserving chosen plaintext attack, and the server is honest-but-curious.

---

> ### Author Rebuttal · Authors · 2023-08-06
>
> [Q1] Homomorphic encryption is used to encrypt labels … high computation overhead … does this impact the efficacy of the proposed method?
>
> [A1] We will clarify that the encrypt labels (by homomorphic encryption) are only used for majority voting in random forests, and it does not require bootstrapping in HE without high computations due to 3-depth homomorphic multiplicative. In contrast, Previous HE work (Wu et al., 2020; Akavia et al., 2022) requires expensive computation costs for bootstrapping during training and prediction.
>
> [Q2] … Table 3, the proposed encrypted random forests perform better than the original random forests on two datasets…explanation…
>
> [A2] We will clarify that our method could achieve better performance as for some noisy datasets, since we encrypt different plaintext features into one ciphertext feature, and this could improve data robustness to noise.
>
> [Q3] Can the proposed Gini-impurity-preserving encryption method be extended to Gini index or information gain?
>
> [A3] We will clarify that our Gini-impurity-preserving encryption can be generalized to other information-theoretic statistics such as Gini index or information gain, and the key problem is to show the piecewise monotonicity of information statistics w.r.t. splitting point.
>
> [Q4] In Algo. 1 , will the selection of c_{max} impact the security of proposed method?
>
> [A4] We will clarify that we select c_{max}=2^{lambda * log_2(# of plaintexts)} with lambda >6.4 as done in (Kerschbaum et al., 2015), while a smaller c_{max} does not guarantee ciphertext’s accuracy within expression range of computer.
>
> We will provide more details on CKKS, Gini-impurity-preserving chosen plaintext attack and bitwise leakage matrices, and improve Figure 6 according to your suggestions. Thank you.

---

> > ### Comment · Reviewer_1ntL · 2023-08-11
> >
> > The rebuttal has solved my concerns.

---

### Official Review · Reviewer_Emcz · 2023-07-09

**Soundness:** 3 good
**Presentation:** 3 good
**Contribution:** 2 fair
**Rating:** 6
**Confidence:** 4

**Summary:**

This work focuses on security and privacy for machine learning, especially random forest and its splitting criterion, Gini impurity. In order to do that, this work proposes a new encryption scheme for features of data, which is against chosen plaintext attack. Moreover, this work utilizes a homomorphic encryption CKKS to encrypt labels of data for secure random forests model.


**Strengths:**

1. In Table 1, this work shows comparisons of asymptotical analysis on training/predictive communication/computation cost and demonstrates their work has advantages over other existing approaches.
2. As shown in Table 3, this work uses a lot of datasets to compare with existing approaches to convince audience on this approach in practice.
3. BST encryption avoids computationally expensive sorting.

**Weaknesses:**

1. Instead of putting proofs in appendix. It is better to use few sentences about how to prove theorems.
2. This work is lack of a related work section to walk through existing work.

**Questions:**

1. In section 4, authors mention [50, 72, 76, 77]. Since they relate to this work, do authors consider to update asymptotical analysis in table 1?

**Limitations:**

1. Chosen plaintext attack somehow is not sufficient in the real world application. If the work can guarantee chosen ciphertext attack, it could be a stronger scheme.

---

> ### Author Rebuttal · Authors · 2023-08-06
>
> [Q1] In section 4, authors mention [50, 72, 76, 77]. Since they relate to this work, do authors consider to update asymptotical analysis in table 1?
>
> [A1] We will clarify that our asymptotical analysis includes [50] in table 1, but does not include [72,76,77], since [76,77] only focuses on prediction without training, while [72] only studies order-preserving scheme without tree training and prediction. We will update relevant asymptotical analysis in table 1 as follows:
>
>
>
> |Scheme|Training |communication | Training comp. |complexity|Predictive  |communication|Predictive comp. |complexity | Model privacy  |
> |  :----: | ----:  |   :---- | ----: |  :----| ----: | :----| ----: |  :---- |:----: |
> |    |  Rounds |  Bandwidth |  Client&emsp;  |  Server |     Rounds | Bandwidth |    Client &emsp;  |  Server |  |
> |[72] |  $×$  &emsp; |  &emsp; $×$ | $×$  &emsp;  |  &emsp;  $×$  | $×$   &emsp; | &emsp;   $×$ | $×$  &emsp; | &emsp;    $×$ |  $×$|
> |[76]|  $×$   &emsp; |  &emsp;  $×$ |  $×$    &emsp;| &emsp;   $×$ |  $O(1)$    &emsp; |  &emsp; $O(1)$   | $O(1) $   &emsp;|  &emsp;   $O(κ) $  |  $√$ |
> |[77] |  $×$   &emsp; |  &emsp;   $×$   |  $×$    &emsp; |  &emsp;  $×$   | $ O(1) $ &emsp; |   &emsp; $O(1) $   | $O(1)$   &emsp; | &emsp;    $O(κ)$   |   $ √ $|
> our  |  $O(h)$  &ensp; | &ensp;  $O(κ\bar{ȷ})$  |  $O(κ) $  &ensp; | &ensp; $O(κ\bar{ȷ}τn)$ |  $O(1)$ &emsp;  |  &emsp;    $O(1) $ | $O(1) $ &emsp;|  &emsp;  $O(h)$    |  $√$ |
>
> Our method takes smaller computational complexity (O(h)) in server since h<κ.
>
> [Q2] Chosen plaintext attack somehow is not sufficient in the real world application. If the work can guarantee chosen ciphertext attack, it could be a stronger scheme.
>
> [A2] We will clarify that our chosen plaintext attack is a tradeoff between security and computation cost, and we can present the chosen ciphertext attack by modifying Alg. 2, motivated by Boldyreva et al. (2012). The basic idea is to introduce additional random perturbations and split each I_i in Alg. 2, whereas this yields expensive computational and space costs for encrypting and training.
>
> We will add a section to introduce relevant work, and present the proof sketches for main theorems according to your suggestions. Thank you.

---

> > ### Comment · Reviewer_Emcz · 2023-08-13
> >
> > Thanks for your table! It looks nice! I think this rebuttal helps!

---

### Official Review · Reviewer_2RPe · 2023-07-10

**Soundness:** 3 good
**Presentation:** 3 good
**Contribution:** 3 good
**Rating:** 6
**Confidence:** 4

**Summary:**

This work focuses on the privacy of random forests. The authors develop a new encryption to preserve the Gini impurity of data, which plays an important role on the construction of random forests. Based on this new encryption, the authors introduce an effective algorithm for privacy random forests under the client-server protocol. Theoretical analysis is presented to guarantee the correctness and security of the proposed algorithm, and extensive experiments validates the effectiveness, efficiency and security of the proposed algorithm. In particularly, the privacy random forests take significantly better performance than previous privacy random forests via encryption, anonymization and differential privacy, and are even comparable to original plaintexts random forests without encryption.

**Strengths:**

Random forests have been one successful algorithm with diverse applications, and this work focuses on the privacy of random forests, which is an important problem and direction in machine learning. Previous methods on this issue take anonymization, differential privacy and homomorphic encryption, to preserve the privacy of random forests, whereas this work explore the data encryption from the crucial ingredients of learning algorithm, i.e., the Gini impurity for random forests.
1.	An interesting encryption with some potential direction: This work proposes an interesting encryption to preserve the minimum Gini impurity, which has been a crucial ingredient on the construction of random forests. The authors also propose an interesting binary search tree for data encryption with O(logn) computational complexity, and it is suitable for real-time or online implementation. This work may motivate us some new direction on data encryption from some crucial ingredients of learning algorithm, and such research could make a bridge between machine learning and encryption.
2.	An effective and efficient privacy-preserved random forests: Based on the encryption of Gini-impurity preservation, the authors present an effective and efficient privacy-preserved random forests under the client-server protocol. As can be seen in Table 1, the proposed random decision tree takes the smallest communication and computational complexities during the training and predictive process of random forests in comparisons with previous privacy-preserving decision trees.
3.	Some theoretical supports for proposed algorithm: The authors prove the preservation of minimum Gini impurity in ciphertexts without decryption, which plays an important role on the construction of random forests. The proposed scheme also satisfies the security against Gini-impurity-preserving chosen plaintext attack.
4.	Good empirical results: Extensive experiments show that the proposed encrypted random forests take significantly better performance than previous privacy random forests via encryption, anonymization and differential privacy, and are comparable to original (plaintexts) random forests without encryption. The authors also show a good balance between computational cost and data security for the proposed algorithm.


**Weaknesses:**


1.	The English presentation should be improved over the whole submission.
2.	The authors could present more detailed analysis on the Gini-impurity-preserving chosen plaintext attack, with necessary background knowledge on similar security threats.
3.	The authors could present the space cost of the proposed encryption algorithm, and it remains unclear whether the algorithm's memory overhead during encryption is significant, especially when dealing with large-scale datasets.


**Questions:**

1. Is the proposed method able to handle multiclass data? Does the number of classes impact the efficiency of the encryption method? If the number of classes is large, could this potentially lead to a sparsity problem?
2. In Algorithm 1, could the selection of the parameter c_max be inappropriate and subsequently affect the ciphertext?
3. I notice that a few methods presented in Table 1 were not included in the experimental comparisons. Could the authors elaborate on this?


**Limitations:**

The authors recognize certain constraints of their proposed method, including the security aspects highlighted in Theorem 3, and the general applicability across varying datasets as evidenced through experiments. However, I encourage the authors to review the proposed method to identify if any limitations persist, specifically those related to computational demands and sensitivity to hyperparameters.

---

> ### Author Rebuttal · Authors · 2023-08-06
>
> [Q1] Is the proposed method able to handle multiclass data? Does the number of classes impact the efficiency of the encryption method? If the number of classes is large, could this potentially lead to a sparsity problem?
>
> [A1] We will clarify that our method could deal with multiclass data efficiently when the number of classes is smaller than 1000, and we could use permutation and compressions techniques for more classes to reduce the sparsity problem.
>
> [Q2] In Algorithm 1, could the selection of the parameter c_max be inappropriate and subsequently affect the ciphertext?
>
> [A2] We will clarify that the parameter c_max=2^{lambda * log_2(# of plaintexts)}, where lambda >6.4 is the security parameter for ciphertexts as in (Kerschbaum et al., 2015). The larger the c_max, the better the security. It is not inappropriate to select smaller c_\max since the ciphertext’s accuracy may exceeds the expression range of computer.
>
> [Q3] … a few methods presented in Table 1 were not included in the experimental comparisons.
>
> [A3] We will clarify that Table 3 summaries the experimental comparisons for the most representative privacy-preserving random forests in recent years, and we omit some comparisons with methods in Table 1 due to pages limitation, as well as relatively weaker performance, which has been shown by previous methods (Aminifar et al., 2021, Akavia et al., 2022).
>
> We will introduce Gini-impurity-preserving chosen plaintext attack with some details, present space cost, and improve this work according to your suggestions. Thank you.

---

> > ### Comment · Area_Chair_U6fz · 2023-08-18
> > **Awaiting Your Feedback on Authors' Rebuttal**
> >
> > Dear Reviewer 2RPe,
> >
> > Thank you for your hard work. The Author-Reviewer discussion ends on August 21. The authors and I are eager to learn whether their responses have adequately addressed your concerns. You are encouraged to directly reply to the authors' rebuttal.
> >
> > Please note that this is a public thread. If you prefer to reply to me individually, please use the internal discussion thread.
> >
> > Kind Regards,
> >
> > AC

---

### Comment · Area_Chair_U6fz · 2023-08-12
**Discussion period**

Dear Reviewers,

I would like to express my sincere gratitude for your thorough examination of this paper. Now that the authors have provided their rebuttal, I kindly ask you to evaluate whether their response sufficiently addresses the concerns you have raised. Should you require any additional information or have further questions, please feel free to request clarification directly from the authors. Your insights and contributions to this process are greatly appreciated!

Best regards,

AC

---

> ### Author Response · Authors · 2023-08-13
>
> Dear Reviewers,
>
> We want to express our gratitudes to your helpful comments and suggestions, which will be of great importance on the improvement of this work.
>
> We have made efforts to address questions you raised and improve accordingly. We would like to double check to make sure that we have addressed all your concerns, in particular for Reviewer PTBU, and would you please let me know if you have any additional questions. Thank you.
>
> Best wishes,
>
> Authors.

---

### Decision · Program_Chairs · 2023-09-21

**Decision:**

Accept (spotlight)

**Comment:**

The paper presents an innovative approach to privacy-preserving random forests by introducing a new encryption scheme that maintains the Gini impurity metric.

The key strengths highlighted by reviewers include that  1). The paper targets pressing challenges in the realm of privacy and machine learning; 2). The proposed encryption scheme and its integration with random forests is a novel contribution; 3). The method's effectiveness is evident both through empirical demonstrations and supported by theoretical findings.

Given the overwhelmingly positive feedback from reviewers and the important niche the paper fills, AC recommends accepting this paper.

To enhance the overall quality of the paper, authors are strongly encouraged to revise their paper according to reviewers' detailed feedback, particularly regarding typographical errors and any unclear claims.